# Epithelial and Mesenchymal-like Pancreatic Cancer Cells Exhibit Different Stem Cell Phenotypes Associated with Different Metastatic Propensities

**DOI:** 10.3390/cancers16040686

**Published:** 2024-02-06

**Authors:** Lisa-Marie Philipp, Umut-Ulas Yesilyurt, Arne Surrow, Axel Künstner, Anne-Sophie Mehdorn, Charlotte Hauser, Jan-Paul Gundlach, Olga Will, Patrick Hoffmann, Lea Stahmer, Sören Franzenburg, Hendrike Knaack, Udo Schumacher, Hauke Busch, Susanne Sebens

**Affiliations:** 1Institute for Experimental Cancer Research, Kiel University, University Hospital Schleswig-Holstein (UKSH), Campus Kiel, 23562 Kiel, Germany; lisa.philipp@email.uni-kiel.de (L.-M.P.); umutyesilyurt97@gmail.com (U.-U.Y.); arne.surrow@me.com (A.S.); stu235744@mail.uni-kiel.de (P.H.); stu213526@mail.uni-kiel.de (L.S.); knaack.hendrike@mh-hannover.de (H.K.); 2Medical Systems Biology Group, Lübeck Institute of Experimental Dermatology, University of Lübeck, 23538 Lübeck, Germany; axel.kuenstner@uni-luebeck.de (A.K.); hauke.busch@uni-luebeck.de (H.B.); 3Institute for Cardiogenetics, University of Lübeck, 23562 Lübeck, Germany; 4Department of General, Visceral-, Thoracic-, Transplantation- and Pediatric Surgery, UKSH, Campus Kiel, 24105 Kiel, Germany; anne-sophie.mehdorn@uksh.de (A.-S.M.); charlotte.hauser@uksh.de (C.H.); jan-paul.gundlach@uksh.de (J.-P.G.); 5Molecular Imaging North Competence Center, Clinic of Radiology and Neuroradiology, Kiel University, UKSH, Campus Kiel, 24118 Kiel, Germany; olga.will@rad.uni-kiel.de; 6Institute of Clinical Molecular Biology, Kiel University, 24118 Kiel, Germany; s.franzenburg@ikmb.uni-kiel.de; 7Academic Affairs Office, Hannover Medical School, 30625 Hannover, Germany; 8Department of Anatomy and Experimental Morphology, University Cancer Center Hamburg, University Medical Center Hamburg-Eppendorf, 20246 Hamburg, Germany; u.schumacher@uke.de

**Keywords:** pancreatic adenocarcinoma, PDAC, cancer stem cells, EMT, epithelial–mesenchymal-transition, heterogeneity, plasticity, adhesion, migration, invasion, metastasis

## Abstract

**Simple Summary:**

Pancreatic ductal adenocarcinoma (PDAC) is characterized by high tumor cell plasticity and heterogeneity, contributing to poor prognosis and treatment failure. Epithelial–mesenchymal transition (EMT) and gain of cancer stem cell (CSC) properties are crucial processes determining tumor cell plasticity. Investigating CSC and non-CSC clones from PDAC cell lines revealed that epithelial and mesenchymal-like CSCs are characterized by different self-renewal abilities and metastatic propensities. Epithelial CSCs were characterized by the expression of the CSC marker SOX2, fast cell growth, and strong self-renewal ability in vitro, together with massive tumor formation in vivo. In contrast, mesenchymal-like CSCs showed a strong expression of the CSC marker Nestin, slower cell growth, self-renewal ability in vitro, and the formation of higher numbers of smaller tumors in vivo. Furthermore, the organ manifestation of mesenchymal-like and epithelial CSC-derived tumors clearly differed. Thus, this study revealed that CSC and non-CSC populations in PDAC can be associated with distinct EMT phenotypes, resulting in distinct metastatic behavior.

**Abstract:**

Pancreatic ductal adenocarcinoma (PDAC) is mostly diagnosed at advanced or even metastasized stages, limiting the prognoses of patients. Metastasis requires high tumor cell plasticity, implying phenotypic switching in response to changing environments. Here, epithelial–mesenchymal transition (EMT), being associated with an increase in cancer stem cell (CSC) properties, and its reversion are important. Since it is poorly understood whether different CSC phenotypes exist along the EMT axis and how these impact malignancy-associated properties, we aimed to characterize CSC populations of epithelial and mesenchymal-like PDAC cells. Single-cell cloning revealed CSC (Holoclone) and non-CSC (Paraclone) clones from the PDAC cell lines Panc1 and Panc89. The Panc1 Holoclone cells showed a mesenchymal-like phenotype, dominated by a high expression of the stemness marker Nestin, while the Panc89 Holoclone cells exhibited a SOX2-dominated epithelial phenotype. The Panc89 Holoclone cells showed enhanced cell growth and a self-renewal capacity but slow cluster-like invasion. Contrarily, the Panc1 Holoclone cells showed slower cell growth and self-renewal ability but were highly invasive. Moreover, cell variants differentially responded to chemotherapy. In vivo, the Panc1 and Panc89 cell variants significantly differed regarding the number and size of metastases, as well as organ manifestation, leading to different survival outcomes. Overall, these data support the existence of different CSC phenotypes along the EMT axis in PDAC, manifesting different metastatic propensities.

## 1. Introduction

Pancreatic ductal adenocarcinoma (PDAC) is the most common malignant pancreatic tumor, characterized by a poor prognosis with 5-year survival rates of around 12% and an ongoing increase in death rates [1,2]. Due to the lack of early and specific symptoms, the majority of patients are diagnosed at an advanced or even metastasized stage, leaving palliative treatment as the only remaining option. The only curative option is surgery; however, only about 20% of patients are eligible for surgery [3,4]. PDAC predominantly metastasizes in the liver, but also in the lungs and peritoneum [4,5,6]. PDAC patients with liver or peritoneal metastases exhibit a significantly shorter disease-free survival rate and overall poorer prognoses than patients with solitary lung metastases [7].

The multistep process of metastasis starts with the dissemination of tumor cells from the primary tumor and ends with the proliferation and outgrowth of macroscopic metastases at secondary sites [8,9,10]. As a prerequisite for dissemination, carcinoma cells have to acquire a motile and invasive phenotype, which is commonly described to be achieved by epithelial–mesenchymal transition (EMT) [11,12]. EMT is associated with the downregulation of adhesion molecules and the loss of epithelial proteins like E-cadherin and conversely, and a gain in mesenchymal characteristics, like the increased expression of mesenchymal markers, such as Vimentin, L1CAM, or the transcription factors ZEB1, ZEB2, and OVOL2 (ZNF339) [11,12,13,14,15,16,17]. Furthermore, this loss of cell differentiation of carcinoma cells has been associated with the acquisition of cancer stem cell (CSC) properties [18,19,20]. Mani et al. were the first who demonstrated that breast cancer cells that have undergone EMT acquire a cancer stem cell-like phenotype, and likewise, stem cell-like cells resemble cells that have undergone EMT [18]. 

CSCs are a small group of cancer cells within a cancer cell population with the unique ability to both self-renew and generate more differentiated cells. Owing to these unique features, CSCs are regarded as essential for tumor initiation at primary and secondary sites, including PDAC [21,22,23]. Recent studies indicate that CSC properties can be gained and lost depending on the microenvironment [20,24,25,26,27], indicating that CSCs are not a stable, but highly plastic cell population. Several markers, e.g., ABCG2, CD133, CD24, CD44, Nanog, Nestin, and SOX2, have been proposed for the identification of CSCs in PDAC, which already indicates a high heterogeneity within the CSC population [28,29,30,31,32,33,34]. The intermediate filament Nestin and the stem cell (SC) transcription factor SOX2 seem to play a role in the maintenance of CSC properties [35,36]. Nestin impacts cell motility and EMT properties in PDAC cells and its knockdown in PDAC cells led to reduced tumor incidence and volume, as well as the formation of liver metastases in a murine PDAC model [37,38]. Elevated SOX2 expression has been rarely detected in pancreatic intraepithelial neoplasia; it is more often detected in poorly differentiated and neurally invasive tumors, supporting the role of this factor in later stages of tumorigenesis and metastasis [39]. In line with these findings, it has been shown that SOX2 is involved in mesenchymal–epithelial-transition (MET), the reversion of EMT; a knockdown of SOX2 in colorectal cancer (CRC) cells altered the expression of key genes involved in the EMT process, including E-cadherin and Vimentin [40]. Furthermore, de novo SOX2 expression in pancreatic cancer cells is sufficient to promote self-renewal and de-differentiation processes, and impact stemness characteristics via modulating specific cell cycle regulatory genes and EMT driver genes [32].

In addition to tumorigenicity assays, it is well established to assess self-renewal abilities of (cancer) cells using the colony-formation assay (CFA), considering not only the number of colonies formed by single cells but also distinct colony morphologies which are associated (Holoclones, Meroclones, and Paraclones) with distinct CSC potentials [41]. While Holoclones are considered to be the colony type with the highest amount of CSCs, Paraclones are characterized by more differentiated non-CSCs [41,42]. Meroclones form an intermediate stage between these two colony types and differ from Holoclones mainly in their lower proportion of CSCs [41,42]. These colony types have been identified in a variety of cancers [42,43,44,45], including PDAC [46]. In accordance with the current understanding that CSCs are plastic cells that are able to gain and lose CSC properties, several studies have demonstrated the phenotypic switching of Holoclones, which differentiate into Mero- or Paraclones [42,46,47,48]. Our group previously demonstrated that Panc1 Holoclone cells, derived using single-cell cloning from the parental PDAC cell line Panc1, showed an elevated and exclusive expression of the CSC marker Nestin compared to Paraclones [41]. 

Of note, experimental evidence has been provided that carcinoma cells with an epithelial phenotype can also contribute to metastasis, using a cluster-like migration pattern, which has also been described for PDAC cells [49]. Moreover, recent studies indicate that so-called hybrid cells concomitantly showing epithelial and mesenchymal characteristics exhibit the highest plasticity regarding cancer stemness, tumor initiation capacity, as well as adaptation capability, which gives these cells the highest probability to metastasize [49]. Still, it is poorly understood whether different CSC phenotypes exist along the EMT/MET axis and how these impact malignancy-associated properties. Thus, to explore whether and how different EMT states are associated with CSC properties in PDAC and how this impacts tumor cell growth and metastasis, Holo- and Paraclone cells were isolated and expanded from another PDAC cell line (Panc89). In contrast to mesenchymal-like Panc1 cells, which have presumably already undergone EMT but are still derived from a primary PDAC, Panc89 cells originate from a lymph node metastasis and have presumably undergone EMT and MET during the metastatic process. Using these well-defined cell models, the present study aimed at a comprehensive in vitro and in vivo analysis of isolated Holo- and Paraclone cell variants from Panc1 and Panc89 cells concerning EMT and CSC properties as well as tumorigenicity and metastasis. Overall, our data support the view of great plasticity and heterogeneity within cancer (stem) cells in PDAC, differentially impacting tumor growth and metastatic behavior.

## 2. Materials and Methods

### 2.1. Cell Lines and Cell Culture

As a model for mesenchymal-like PDAC cells, the human cell line Panc1 was used (purchased from ATCC, Manassas, VA, USA), originating from a primary tumor of a PDAC patient. As a model for epithelial PDAC cells, the human cell line Panc89 (kindly provided by Prof. T. Okabe, University of Tokyo, Tokyo, Japan), originating from a lymph node metastasis of PDAC, was used. Holo- and Paraclone cells of both cell lines were isolated and expanded via single-cell cloning (see below). All cell lines were cultivated in Panc-medium (RPMI 1640 supplemented with 10% FCS, 1% L-glutamine, and 1% sodium pyruvate (Biochrom, Berlin, Germany)).

For analysis of PDAC cell adhesion behavior to endothelial cells of different metastatic sites, lung (HuLEC-5a) and liver (TMNK-1) endothelial cell lines as well as the mesothelial cell line Met-5a were used. HuLEC-5a cells (purchased from ATCC, Manassas, VA, USA) are human microvascular endothelial cells that are derived from the lungs of a male patient [50]. They were cultured in MCDB 131 (Biochrom, Berlin, Germany) medium containing 10% FCS, 1% L-glutamine, 10 ng/mL epithelial growth factor and 1 µg/mL hydrocortisone. TMNK-1 (purchased from NIBIOHN JCRB cell bank, Osaka, Japan) are human liver sinusoidal endothelial cells [51], originating from the liver of a female patient [51]. They were cultured in DMEM (Biochrom, Berlin, Germany) containing 10% FCS, 1% L-glutamine, and 1% penicillin–streptomycin. Met-5a cells (purchased from ATCC, Manassas, VA, USA) originating from mesothelium, were isolated from the pleural fluids of non-cancerous individuals [52,53] and cultured in Medium 199 (Sigma Aldrich/Merck KGaA, Darmstadt, Germany) containing 10% FCS, 400 nM hydrocortisone, 870 nM porcine insulin, and 20 nM HEPES.

### 2.2. Single-Cell Cloning and Clone Expansion

Single-cell cloning and expansion were performed with parental PDAC cell lines to isolate and expand single-cell-derived Holo- and Paraclone cells. Single-cell cloning of Panc1 cells to isolate Panc1 Holo- and Paraclone cells, respectively, was performed previously [41]. To isolate Holo- and Paraclone cells from Panc89 cells, the parental cells were pre-diluted to a cell count of 1000 cells per mL Panc-medium. From this cell suspension, 100 µL (corresponding to 100 cells) were added to 20 mL of Panc-medium to obtain a concentration of one cell per 200 µL of cell medium. Cells were seeded 1 cell/well in a transparent, flat 96-well plate. The plate was centrifuged, and the first screening was performed directly after centrifugation by imaging the plate in the brightfield channel of a NYONE^®^ Scientific Imager (SYNENTEC GmbH, Elmshorn, Germany) and the brightfield channel of the EVOS microscope (AMG, Bothell, WA, USA). Only wells containing exactly one cell were considered for further monitoring and expansion. Microscopical screening was performed twice per week. After 12–20 days, the colony shape was determined, and the colonies were monitored until a cell confluence of about 80% was reached. Cell clones with a stable colony morphology [41,42] were detached from one well and transferred to one well of a 6-well plate. The morphology of expanded cell clones was checked regularly. Only clones with a clear CSC or non-CSC morphology [41,42] and a stable phenotype were further cultivated, expanded, and used for phenotypic and functional characterization. Phenotypes were regularly checked using a marker analysis via qPCR. All experiments of this study were performed with defined Holo- and Paraclone cells isolated from both Panc1 cell clones (clone 9, clone5, respectively) and Panc89 cell clones (clone 4, clone 3, respectively). 

### 2.3. Colony-Formation Assay (CFA)

The self-renewal ability of different cell variants was analyzed using CFA. According to established protocols [41,46,48], a low number of 400 cells was seeded as duplicates or triplicates in 6-well plates in Panc-medium. After cultivation for 8–11 days, colonies were fixed with 4% Paraformaldehyde (PFA, Thermo Scientific, Schwerte, Germany) for 10 min, stained with 0.1% crystal violet (Merck Millipore, Darmstadt, Germany) for 1 h, washed in dH_2_O and air-dried at RT. Only colonies containing more than 50 cells were counted and their morphology regarding Holo- and Paraclone was determined. Holoclones consisted of tightly, homogenously clustered cells with a regular borderline, while Paraclones consisted of dispersed, larger cells with an irregular boundary [46]. Meroclones, defined as an intermediate colony type [46], were neglected in this study. 

### 2.4. RNA Isolation and RT-qPCR

Total RNA was isolated using the total RNA kit peqGOLD (PeqLab, Erlangen, Germany) and subjected to reverse transcription according to the manufacturer’s instructions (Fermentas via Thermo Fisher Scientific, Darmstadt, Germany). The qPCR analysis was performed in duplicates on a LightCycler 480 (Roche, Basel, Switzerland) for a maximum of 50–60 cycles ending with a melting curve analysis for primer quality control. Primers (Eurofins, Ebersberg, Germany; RealTime Primers via Biomol, Hamburg, Germany), primer sequences, and annealing temperatures that were used are listed in Table 1. For relative quantification of RNA levels, C_T_ values of genes of interest were normalized to the respective C_T_ value for the reference gene GAPDH.

### 2.5. Analysis of CSC and EMT Markers via Immunofluorescence Staining

For the analysis of different CSC and EMT markers on the protein level in Panc1 and Panc89 cell variants, immunofluorescent staining (IFS) was performed. Panc1 cell variants were stained with anti-Nestin (clone 10C2; Thermo Scientific, Schwerte, Germany) and anti-ZEB2 (polyclonal; Novus Biologicals, Wiesbaden Nordenstadt, Germany) antibodies, while all Panc89 cell variants were stained with anti-SOX2 (clone D6D9; Cell Signaling Technology, Danvers, MA, USA) and anti-L1CAM (clone UJ127.11; Gerd Moldenhauer, German Cancer Research Center, Heidelberg, Germany) antibodies.

First, glass coverslips were placed in each well of a 12-well plate. Then, 0.5 × 10^4^ cells/well were seeded. After 48 h, cell culture medium was removed and the cells were washed with PBS. Fixation of the cells was performed by incubation with 4% PFA for 15 min at RT. Next, the coverslips were washed for 3 × 5 min with PBS and incubated with ice-cold methanol (MetOH) for 10 min at −20 °C for permeabilization of the cells. Afterward, the coverslips were washed with PBS and cells were blocked with 4% bovine serum albumin (BSA)—0.3% TritonX-100 in PBS for 1 h at RT. After a washing step with 0.3% TritonX-100/PBS, incubation with the primary antibodies was performed (see below).

#### 2.5.1. Concomitant Double IFS of Nestin and ZEB2 in Panc1 Cells

Incubation of the antibodies and Hoechst 33258 (Merck Millipore, Darmstadt, Germany) was carried out in 1% BSA–0.3% TritonX-100/PBS and in a humidity chamber. Incubation of the primary antibodies was performed at 4 °C overnight (ON). Anti-Nestin antibody (Thermo Scientific, Schwerte, Germany) was diluted 1:200 (=5 µg/mL); anti-ZEB2 antibody (Novus Biologicals, Wiesbaden-Nordenstadt, Germany) was diluted 1:50 (=2 µg/mL); mouse IgG1 isotype control (R&D Systems GmbH, Wiesbaden, Germany) was diluted 1:100 (=5 µg/mL); and rabbit IgG isotype control (Bio-Techne, Minneapolis, MN, USA) was diluted 1:500 (=2 µg/mL). Incubation of primary antibodies for concomitant double staining was performed by applying both primary antibodies at the same time. After ON incubation, cells were washed with 0.3% TritonX-100/PBS. Afterward, anti-mouse antibodies conjugated with AlexaFluor 488 (Invitrogen, Carlsbad, CA, USA), anti-rabbit antibodies conjugated with AlexaFluor 647 (Invitrogen, Carlsbad, CA, USA) and Hoechst 33258 were each diluted 1:500 in 1% BSA–0.3% TritonX-100/PBS and added to the cells for 1 h at RT. After washing with PBS and dH_2_O, coverslips were sealed with Fluor Safe Reagent (Electron Microscopy Sciences, Hatfield, Panama) and transparent nail polish.

#### 2.5.2. Sequential IFS of SOX2 and L1CAM in Panc89

Incubation of anti-SOX2 (Cell Signaling Technology, Danvers, MA, USA) and the related secondary antibody was carried out in 1% BSA–0.3% TritonX-100/PBS and in a humidity chamber. Incubation of anti-L1CAM and the related secondary antibody, as well as Hoechst 33,258 staining, was carried out in 1% BSA/PBS. Anti-SOX2 antibody was diluted 1:200 (=25 µg/mL); anti-L1CAM antibody was diluted 1:100 (=10 µg/mL); mouse IgG1 isotype control was diluted 1:50 (=10 µg/mL); and rabbit IgG isotype control was diluted 1:40 (=25 µg/mL). Incubation of primary antibodies was performed sequentially, starting with the ON staining of SOX2 at 4 °C. After incubation with an anti-SOX2 antibody, cells were washed with 0.3% TritonX-100/PBS. Incubation with anti-rabbit antibody, conjugated with AlexaFluor 647 and diluted 1:500 in 1% BSA–0.3% TritonX-100/PBS, was performed for 1 h at RT. After washing with PBS, incubation with anti-L1CAM antibody was carried out for 1 h at RT. After washing with PBS, cells were incubated with the anti-mouse antibody conjugated with AlexaFluor 488 and Hoechst 33258, both diluted 1:500 in 1% BSA/PBS for 1 h at RT. After washing with PBS and dH_2_O, coverslips were sealed with Fluor Safe Reagent and transparent nail polish. Image acquisition and staining evaluation of all cell lines were performed with the Lionheart FX Automated Microscope and related software (Gen5 Data Analysis Software 3.10 (BioTek, Bad Friedrichshall, Germany).

### 2.6. RNA Sequencing and Transcriptomic Analysis

Total high-purity RNA of all cell variants was isolated each from three different cell passages using the total RNA kit peqGOLD (PeqLab, Erlangen, Germany) with an additional DNase digestion step according to manufacturer’s specifications. Afterwards, the RNA triplicates of each cell line were placed in a 96-well plate in a concentration of at least 150 ng/15 µL RNase-free water. RNA sequencing was performed at the Institute of Clinical Molecular Biology (Kiel, Germany) under the supervision of Sören Franzenburg. Input total RNA was quantified using the Quant-it RNA Assay (Thermo Fisher, Waltham, MA, USA), and RIN scores were determined using the Agilent Tape Station (Agilent, Santa Clara, CA, USA). All RIN scores were >8.200 ng. Input total RNA was processed using the Illumina stranded mRNA kit (Illumina, San Diego, CA, USA). Libraries were sequenced on the Illumina NovaSeq 6000 SP Flowcell using 100 bp paired-end reads. Transcriptomic analysis of mRNA sequencing data was performed by Hauke Busch and Axel Künstner (Medical Systems Biology Group, Institute of Experimental Dermatology, University of Lübeck, Germany). Raw sequencing data (fastq format) were mapped against the human transcriptome (Ensembl GRCh38.106) using kallisto (v0.46.1) [54], and gene expression was summarized from the scaled TPM values of the transcripts using tximport [55]. Differential gene expression was performed using DESeq2 [56]. Gene set enrichment analysis (GSEA) on the shrunken log2 fold changes from apeglm [57] was performed using GAGE (v2.48.0) [58] against HALLMARK, REACTOME and Gene Ontology Biological Processes (GOBP) gene sets, extracted from the msigdbr R package (v7.5.1). Gene sets from the Molecular Signatures Database collection C2 [59], selected for Reactome pathways and a stemness gene set “MUELLER PLURINET”, were evaluated via GSEA.

### 2.7. Cell Growth Analysis

Panc1 and Panc89 cell variants were seeded at 5 × 10^3^ cells/well in triplicates in 96-well plates for seven days (168 h). Cells were imaged every day with the brightfield channel (confluence operator) and the UV channel (nuclei count operator) of the NYONE^®^ Scientific Imager (SYNENTEC GmbH, Elmshorn, Germany). To analyze the number of nuclei, every day three wells per each cell variant were dyed using Hoechst 33342 staining (1:5000; Thermo Fisher, Carlsbad, CA, USA). The examined data were analyzed via nonlinear regression (curve fit) to maintain the growth rate (κ) for logistic growth for the nuclei count analysis (k_Nuclei count_) (YT-SOFTWARE^®^ 22.02.24564 (SYNENTEC GmbH, Elmshorn, Germany) in GraphPad Prism 9.5.0 (GraphPad Software, La Jolla, CA, USA).

### 2.8. Treatment Response Analysis

To assess treatment responses of different PDAC cell variants, the total number of adherent cells, detected after being left untreated or exposed to treatment with cytostatic drugs for 72 h, was determined. For this purpose, Panc1 and Panc89 cell variants were seeded at 5 × 10^4^ cells/well in duplicates in 96-well plates. After 24 h, cells were left untreated for control or treated with 0.0038 µM of the standard cytostatic drug Gemcitabine. After 72 h, cells were washed with fresh cell culture medium, stained with Hoechst 33342 (1:5000; Thermo Fisher, Carlsbad, CA, USA), and nuclei of the adherent cells were counted using the NYONE^®^ Imager (YT-SOFTWARE^®^ 22.02.24564 (SYNENTEC GmbH, Elmshorn, Germany). Nuclei count data of treated cells were normalized to untreated cells. 

### 2.9. Migration Assay

To analyze cell migration of Panc1 and Panc89 cell variants, two-chambered silicone inserts by Ibidi^®^ (Ibidi GmbH, Martinsried, Germany) with a defined cell-free gap were used. Before use, inserts were pre-warmed at 37 °C for about 30 min and then placed into wells of a 24-well plate. Per each chamber of an Ibidi^®^ insert, 5 × 10^4^ PDAC cells were seeded in a volume of 75–100 µL Panc-medium and incubated for at least 24 h or to a cell confluence of approximately 100%. Afterward, inserts were removed, resulting in a cell-free gap between the cell layer of both chambers and wells were refilled with 1 mL of FCS-free Panc-medium. Immediately after removal of the inserts, the wells were imaged using CELLAVISTA^®^ or NYONE^®^ (SYNENTEC GmbH, Elmshorn, Germany) for 8 h. The wound-healing application of YT-SOFTWARE^®^ 22.02.24564 (SYNENTEC GmbH, Elmshorn, Germany) automatically determined the cell-free gap and analyzed the confluence in this initial gap. As some cells/debris remained in the gap, the initial confluence was up to 20%, which was subtracted from all time points.

### 2.10. Invasion Assay

The invasive behavior of Panc1 and Panc89 cell variants was analyzed in spheroid invasion assays. For this purpose, 5 × 10^3^ cells/well were seeded in 96-well ultra-low-attachment plates (Corning, Kennebunk, ME, USA; faCellitate, Mannheim, Germany) in 200 µL Panc-medium, enabling spheroid formation. Directly after seeding, the plate was centrifuged. After spheroid formation for 48 h, 150 µL of cell Panc-medium was removed from each well and replaced by 50 µL Matrigel (Corning, New York, NY, USA) to achieve a protein concentration of 6.25 mg/L. After the addition of Matrigel, the plate was centrifuged and incubated at 37 °C, 5% CO_2_, and 85% humidity. After 30 min, the first imaging was performed using the NYONE^®^ Scientific imaging device and the YT^®^ *Spheroid invasion* operator. The duration for monitoring cell invasion was chosen according to the invasive behavior of the PDAC cell variants. To account for these differences, Panc1 cell variants were imaged at time points 0, 24, 48, 72 h, while Panc89 cell variants were measured at time point 0 h as well as after 5 and 7 days. The determination of invasive fronts was carried by counting the number of protruding cells or cell clusters in each spheroid in the image data generated by the NYONE^®^ Scientific Imager (YT-SOFTWARE^®^ 22.02.24564, SYNENTEC GmbH, Elmshorn, Germany). Invasive distances were measured using a digital measuring instrument integrated into the YT-SOFTWARE^®^ 22.02.24564. Specifically, the invasive distance was determined by utilizing a green-marked circle corresponding to the spheroid diameter as the starting point, while the distal edge of the cells facing away from the spheroid was designated as the end point of maximum invasion.

### 2.11. Adhesion Assay

To analyze cell adhesion of Panc1 and Panc89 cell variants, 1 × 10^5^/well HuLEC-5a, Met-5a, or TMNK-1 cells, each diluted in 200 µL of their respective medium, were seeded in a 96-well plate. After 24 h, Panc1 and Panc89 cell variants were added. Before they were added, 5 × 10^5^ cells/mL PDAC cells were suspended in FCS-free Panc-medium and stained with CellTracker Green CMFDA (5-chloromethylfluorescein diacetate, diluted 1:2000, Thermo Scientific, Schwerte, Germany) for 30 min at 37 °C under consistent agitation.

After staining, cells were washed with Panc-medium and seeded at 1 × 10^4^ cells/well in 200 µL Panc-medium onto HuLEC-5a, Met-5a or TMNK-1 cells. Afterward, plates were centrifuged at 300× *g* for 5 min at RT to ensure subsequent contact of all PDAC cells to endothelial or mesothelial cells. Immediately thereafter, the plate was measured in the NYONE^®^ Imager (SYNENTEC GmbH, Elmshorn, Germany), determining the maximal fluorescence count of PDAC cells (*t* = 0 h). To determine the fluorescence of the PDAC cells that adhered to the endothelial or mesenchymal cells, measurement was performed after 4 h. Before measuring, a medium containing non-adhered PDAC cells was removed from the wells and replaced with fresh Panc-medium. The number of fluorescent cells at 4 h was compared to the number of fluorescent cells at 0 h to determine the adhesion rate. The adhesion rate was calculated as the percentage of adherent cells by dividing the number of fluorescent cells at each time point by the number of fluorescent cells at 0 h.

### 2.12. Tumorigenicity and Metastasis Assay In Vivo

All animal studies were executed in compliance with the European guidelines for care and use of laboratory animals and approved by local authorities (V242-44598/2018 (80-8/18)). The tumorigenicity and metastasis assay for Panc1 Holo- and Paraclone cells was already performed [41]. The analysis of tumorigenicity and metastatic behavior of Panc89 Holo- and Paraclone cells was performed accordingly [41]. For this purpose, 1 × 10^4^ cells of either Panc89 Holo- or Paraclone cells were diluted in 75 µL PBS and intrasplenically inoculated in 8-week-old, female SCID beige mice (each group *n* = 10) (Charles River, Sulzfeld, Germany). Progression of tumor formation was monitored regularly using palpation and abdominal ultrasound examination using Vevo 770 (FUJIFILM VisualSonics Inc., Toronto, ON, Canada). Mice inoculated with Panc89 cell variants were sacrificed when the health status was impaired due to a high tumor burden that demanded the removal of the animals from the experiment. In contrast, animals inoculated with Panc1 cell variants showed very slow tumor progression, leading to only small lesions which did not further grow out even after 5 months (=146 days). Therefore, it was decided to terminate the experiment prematurely. Organs and tumors were fixed in 4% PBS-buffered formalin and embedded in Paraffin for sectioning and immunohistological examination.

### 2.13. Immunohistochemical Staining of Paraffin-Embedded Tissue Sections

Formalin-fixed and Paraffin-embedded (FFPE) organs and tumors from Panc1 or Panc89 Holo- or Paraclone cell-inoculated mice were used for immunohistochemical (IHC) analysis. Antigen retrieval was performed as listed in Table 2 and tissue sections were stained with antibodies listed in Table 3. Incubation of anti-PanCK, anti-L1CAM, anti-ZEB1, anti-Nestin, and anti-SOX2 was performed for 1 h at RT (30 min for anti-E-cadherin), followed by 30 min incubation at RT for all secondary antibodies (15 min for ZEB2 secondary antibody).

IHC sections were digitalized (Axioscan Z1, Carl Zeiss AG, Jena, Germany) and stainings were analyzed using ZEN3.3 software (Carl Zeiss AG, Jena, Germany). To quantify IHC staining of tumors, a tumor score was defined to characterize tumors in terms of percentage of positive stained cells (Frequency) and staining intensity (Intensity). The frequency score was set from 1–4, and the intensity score was set from 1–3 (Table 4).

### 2.14. Statistical Analysis

The statistical evaluation was carried out using GraphPad Prism (Version 9.5.0 (GraphPad Software by Docmatics, San Diego, CA, USA). All data sets were tested for normal distribution using the Shapiro–Wilk test.

Parametric data including multiple groups were tested using one-way analysis of variance (one-way ANOVA) for statistical significance. Non-parametrical datasets of multiple groups were analyzed with Kruskal–Wallis one-way ANOVA on ranks test. Statistically significant differences between the groups were assumed at *p*-values ≤ 0.05 according to the Student–Newman–Keuls method (parametric data) and Dunn’s method (non-Parametric data), respectively.

Graphs of parametric data were presented as mean with standard deviation or mean with standard error of means (depending on technical and biological replicates), while graphs of non-parametric data were presented as median with (interquartile) range. Student’s *t*-test was used to examine two samples of normally distributed data.

Columned data sets were analyzed using one-way ANOVA plus Tukey´s multiple comparisons test for parametric data and Kruskal–Wallis test plus Dunn´s multiple comparisons test for non-parametric data. Grouped data sets were analyzed using two-way ANOVA and Tukey´s multiple comparisons test. Growth curve experiments were analyzed via nonlinear regression (curve fit) to maintain the growth rate (κ) for logistic growth.

Survival analysis was performed using Kaplan–Meier–Survival analysis with curve comparison and Mantel–Cox logrank test, resulting in significantly different survival curves with a *p*-value of ≤0.001 (***).

Statistical significance was defined at a *p*-value of ≤0.033. Significances are indicated by asterisks: *p* ≤ 0.033 = *, *p* ≤ 0.002 = **, *p* ≤ 0.001 = ***.

## 3. Results

### 3.1. In Vitro Analysis of Panc1 and Panc89 Cell Variants

#### 3.1.1. Panc1 and Panc89 Cell Variants Exhibit Differences in Colony-Formation Ability

From the two parental PDAC cell lines (Panc1 and Panc89), each representing a heterogeneous cell population with different cell variants, Holo- and Paraclone cell lines were generated via single-cell cloning. First, we comparatively analyzed the colony-formation ability of the parental Panc1 and Panc89 as well as Holo- and Paraclone cell variants, which is indicative of the cellular self-renewal properties. Figure 1A depicts typical colony morphologies of Holoclones and Paraclones in the three cell variants of each PDAC cell line.

Regarding the total number of colonies (Figure 1B), the parental Panc1 and Panc89 cells formed a comparable number of colonies (12 and 11, respectively). In contrast, the Panc89 Holo- and Paraclone cells (102 and 51, respectively) formed considerably more colonies compared to the respective Panc1 cell counterparts (9 and 11, respectively). Furthermore, the parental Panc1 cells formed more Paraclone than Holoclone colonies, while the parental Panc89 cells gave rise to more Holoclone colonies, suggesting the prevalence of Paraclones in the parental Panc1 cells and Holoclones in the parental Panc89 cells. Albeit the Panc1 Holoclone and Paraclone cells formed a similar number of colonies, both cell variants clearly differed with respect to the colony type. In line with their clonal origin, the Panc1 Holoclone cells predominantly formed Holoclone colonies and the Panc1 Paraclones predominantly formed Paraclone colonies. The Panc89 Holoclone cells also predominantly gave rise to Holoclone colonies, which was also clearly more pronounced compared to the Panc89 Paraclone cells. Although the latter cells formed considerably more Paraclone colonies than Panc89 Holoclones, these cells gave rise to almost comparable numbers of Holoclone and Paraclone colonies. Thus, the Panc89 Holoclone cells most strikingly differed from the Paraclone cells regarding their ability to generate more colonies in general, and in particular, of colonies with a Holoclone morphology. 

Overall, these findings indicate a high heterogeneity with respect to the self-renewal capacity between the parental PDAC cell lines as well as between respective Holo- and Paraclone cells.

#### 3.1.2. Parental Panc1 and Panc89 as Well as Their Derived Holo- and Paraclone Cells Show Distinct Transcriptional CSC and EMT Signatures

To further comparatively characterize the different PDAC cell variants, a transcriptomic analysis was performed, revealing that, in total, 10,429 out of 17,354 detected genes were differentially regulated in the parental Panc1 and Panc89 cells. Furthermore, a Principal Component Analysis (PCA) on the 10% most variable transcripts of all samples aligned the two PDAC cell lines along the highest principal component (PC1) and aligned the cell variants orthogonally along the second highest principal component (PC2), with a variance of about 77% and 7%, respectively (Figure 2A). These results illustrate clear transcriptional differences between the two PDAC cell lines, Panc1 and Panc89, but also between the respective Holo- and Paraclone variants of each cell line. To further elucidate these differences between cell variants, especially between Holo- and Paraclone cell populations, gene expression data were comparatively evaluated.

A pathway enrichment analysis of the sequenced RNA samples from the parental Panc1 and Panc89, Holo- and Paraclone cells was performed in order to compare Reactome and stemness pathway activities (Appendix A, [57,58]). The comparison of parental cell lines (Appendix A) as well as the comparison of the Holo- and Paraclones of Panc1 cells (Appendix A) and Panc89 cells (Appendix A) demonstrated clearly differing pathway activities, e.g., related to processes like stemness, cell growth and division, or cell extracellular matrix interaction.

Focusing on EMT, invasion, and cancer stemness, the GSEA revealed distinct differences between both parental Panc1 and Panc89 cells, but also between their derived Holo- and Paraclone cell variants. The parental Panc1 cells showed a positive regulation of invasion and increased EMT activity compared to the Panc89 cells, while a stronger stemness potential was determined in the Panc89 cells compared to the Panc1 cells (Figure 2B). Comparing the Holo- and Paraclones of the respective cell lines, it could be shown that the Panc89 Holoclones significantly exhibited a higher stemness potential, together with an increased regulation of invasion compared to the Panc89 Paraclone cells, which presented significantly more EMT activity (Figure 2B). Similarly, the Panc1 Holoclone cells showed a significantly higher stemness potential compared to the Panc1 Paraclones, which were characterized by significantly more EMT activation (Figure 2B).

Furthermore, a GSEA performed on the marker genes already described for PDAC subtypes according to Moffitt et al. [60] further supported clear differences in the transcriptomic signature between parental Panc1 and Panc89 cell lines (Appendix A), but also between their clone variants (Appendix A). Here, only faint expression differences regarding genes associated with the classical-like subtype could be noted in both PDAC cell lines, while the parental Panc89 cells exhibited a clearly enhanced gene signature of the basal-like subtype compared to the parental Panc1 cells (Appendix A). Furthermore, comparing these signatures in the Holo- and Paraclones of Panc1 as well as of the Panc89 cells confirmed the specific differences between the cell lines, but also revealed clearly divergent gene expressions between the respective clone variants (Appendix A). 

A more detailed analysis of the gene expression of EMT and CSC-related genes further substantiated differences between the two PDAC cell lines as well as between the respective Holo- and Paraclone cells (Figure 2C). In the Panc1 cells, a higher expression of the CSC marker *NES* as well as the EMT-associated genes *VIM*, *L1CAM*, and *ZEB1* was observed, while the Panc89 cells displayed a higher expression of the CSC markers SOX2 and OVOL2, as well as the epithelial marker *CDH1* (E-cadherin) (Figure 2C).

While all the Panc1 cell variants expressed similar levels of the mesenchymal markers *VIM*, *L1CAM*, and *ZEB1*, clear differences in the expression of the CSC marker *NES* were observed. Here, the parental Panc1 and Panc1 Holoclone cells showed a higher *NES* expression compared to the Panc1 Paraclones (Figure 2C), indicating this gene as a reliable marker to discriminate CSC (Holoclone) from non-CSC (Paraclone) populations of Panc1 cells. All the Panc89 cells exhibited similar levels of the epithelial marker *CDH1*; however, the expression of the other genes markedly differed among the distinct cell variants. While the parental Panc89 and Panc89 Holoclone cells were characterized by a higher expression of *SOX2* compared to the Panc89 Paraclone cells, the expression of *OVOL2* and *L1CAM* was lowest in the Panc89 Holoclone cells compared to the other two Panc89 cell variants. Additionally, the Panc89 Paraclones expressed higher levels of the mesenchymal marker *VIM* and EMT inducers *ZEB1* and *ZEB2* compared to the parental Panc89 and Holoclone cells (Figure 2C). Overall, the most striking expression differences in the Panc89 Holo- and Paraclones were noted regarding *SOX2, OVOL2*, and *L1CAM*.

To validate the transcriptomic signatures identified with RNA sequencing, a qPCR analysis of CSC markers (*NES*, *SOX2*, and *OVOL2*) and EMT markers (*CDH1*, *L1CAM*, *VIM*, *ZEB1*, and *ZEB2*) was performed on the different PDAC cell variants. 

As shown in Figure 2, differences in the RNA expression of the CSC markers *NES*, *SOX2*, and *OVOL2*, as well as of the EMT markers *CDH1*, *L1CAM*, *VIM*, *ZEB1*, and *ZEB2* could be confirmed (Figure 2D–F). Furthermore, this analysis revealed that Panc1 Holoclone cells (=CSC population) exhibited clearly higher *NES* and *ZEB2* expression levels compared to Panc1 Paraclone cells (=non-CSC population). In contrast, the Panc89 Holoclone cells (=CSC population) could be discriminated from the Panc89 Paraclone cells (=non-CSC population) by higher *SOX2* expression levels in the absence of *L1CAM* expression.

Finally, double IFS was performed to confirm the gene expression differences on protein levels and particularly to discriminate Holo- and Paraclone cells from each other. Since the Panc1 Holo- and Paraclone cells differed with respect to *NES* and *ZEB2* expression, and the Panc89 Holo- and Paraclone cells differed regarding the expression of *SOX2* and *L1CAM*, these markers were stained during IFS.

The parental Panc1 cell population comprised cells exhibiting both Nestin and ZEB2 levels, but also cells lacking both markers, supporting the view of a heterogeneous cell population consisting of CSCs and non-CSCs (Figure 2G). In line with the gene expression status, clear differences were determined between the Panc1 Holo- and Paraclone cells, with the Panc1 Holoclone cells being clearly positive for Nestin and ZEB2, while the Panc1 Paraclone cells lacked the expression of both markers (Figure 2G). An analysis of SOX2 and L1CAM in the parental Panc89 cells revealed high levels of both markers but showed no co-expression pattern of SOX2 and L1CAM within the heterogeneous population, as the colonies that formed were either positive for SOX2 or L1CAM. In line with the gene expression, the Panc89 Holoclone cells were SOX2 positive but lacked L1CAM expression, while the Panc89 Paraclone cells exhibited strong L1CAM expression while being SOX2 negative (Figure 2G).

Overall, these data indicate the existence of different CSC phenotypes which can be associated either with a mesenchymal-like or an epithelial phenotype. While the Panc1 Holoclone cells are characterized by a mesenchymal-like stemness phenotype with a pronounced expression of the CSC marker Nestin, the Panc89 Holoclone cells are characterized by an epithelial SOX2-dominated stemness phenotype.

#### 3.1.3. Panc89 Cell Variants Show Enhanced Cell Growth Rates Compared to Panc1 Cell Populations and Differ with Respect to Responses to Chemotherapeutic Treatments

Having identified different CSC–EMT phenotypes (Holoclones), we next analyzed their functional behavior in comparison to their parental and Paraclone populations. First, the cell growth of the Panc1 and Panc89 cell variants was analyzed by determining nuclei counts over time (168 h). 

In general, all the epithelial Panc89 cell variants grew clearly faster compared to the mesenchymal-like Panc1 cell variants, which was also indicated by the higher k_Nuclei count_ values (Figure 3A).

Moreover, the Holoclone cells of either cell line grew faster compared to the respective Paraclone cells, so that the growth of the Panc1 Holoclones was increased by 29% compared to the Panc1 Paraclone cells. The Panc89 Holoclone cells grew 30.65% faster than the Panc89 Paraclones. Overall, these data underscore that mesenchymal-like PDAC cells exhibit a slowly growing phenotype compared to epithelial PDAC cells, which also applies to their CSC fraction.

Next, it was analyzed whether these different cell-growth abilities influence the response to cytostatic drugs. For this purpose, the Panc1 and Panc89 cell variants were either left untreated or treated with the cytostatic drug Gemcitabine for 72 h. In line with the slower growth behavior, all Panc1 cell variants showed a poorer response towards Gemcitabine compared to the Panc89 cell variants, indicated by the higher percentage of nuclei of still adherent cells (Figure 3B). Thus, Gemcitabine treatment led to a reduction in cell numbers of 70–90% in the Panc89 cell variants but only 40–60% in Panc1 cell variants. Of note, distinct cell clone-dependent variances could be determined. While Panc1 Holoclone cells showed the poorest drug response, the Panc89 Holoclone cells significantly exhibited the strongest reduction in cell number after treatment with Gemcitabine. Altogether, these data indicate a poorer treatment response of mesenchymal-like Panc1 cell variants compared to the epithelial Panc89 cell variants, which particularly manifested in the CSC population and which correlated with the differences in growth behavior. Thus, these findings suggest that mesenchymal-like CSCs seemed to be even more drug-resistant compared to their parental or non-CSC variants, while epithelial CSCs responded even better compared to their parental or non-CSCs variants, further underscoring the heterogeneity among CSC populations. 

#### 3.1.4. Panc1 Holoclone Cells Are Less Migratory but Highly Invasive in a Mesenchymal-like Invasion Manner, While Panc89 Holoclone Cells Show Pronounced Cell Migration but Slow Invasion in Clusters

We next analyzed the migratory behavior of the parental Panc1 and Panc89 and their derived Holo- and Paraclone cells. After 8 h, the cell confluence on the gap was less than 20% for all Panc1 cells, with no differences between cell variants (Figure 4A). In contrast, the cell confluence of all the Panc89 cell variants was much higher, with the Panc89 Holoclone cells leading to the highest cell confluence (75%) on the gap. The Panc89 Paraclone cells showed only 25% cell confluence, while the parental Panc89 cells showed about 35% cell confluence on the gap, indicating that the epithelial Panc89 cells migrated faster than the mesenchymal-like Panc1 cells; particularly, the epithelial CSC fraction exhibited the highest migration potential (Figure 4A). Besides these clear differences in the migration velocity, the Panc1 and Panc89 cells used different modes of migration, fitting together with the different EMT phenotypes. While the Panc89 Holoclone cells migrated in cell clusters with many cell–cell contacts, the Panc1 Holoclone cells exhibited a mesenchymal-like migration of single cells (Figure 4B).

To analyze invasive properties, all PDAC cell lines were seeded in ULA plates to form spheroids. Initial tests revealed that the cell invasion of Panc1 cells is much faster than those of the Panc89 cell variants, resulting in a monitoring time of 3 days for the Panc1 cell variants and 7 days for the Panc89 cell variants, respectively. Accordingly, the Panc1 cell populations formed more aggregate-like spheroids, indicating lower amounts of tight cell–cell contacts, while spheroids formed by either of the Panc89 cell variants were compact and tightly-packed, leading to smaller spheroids compared to those of Panc1 cell populations (Figure 4C). Spheroid sizes of all Panc1 cell variants were about 1000 µm at day 0 and did not considerably change until day 3. In contrast, all Panc89 cell variants formed spheroids of about 300 µm on day 0 which further increased in size up to 500 µm at day 7 (Figure 4C,D). Of note, the mode of cell invasion clearly differed between the Panc1 and Panc89 cells. While the Panc1 cell variants showed single cell invasion, all Panc89 cell variants invaded as clusters, comparable to tumor cell buds [61]. Moreover, huge differences in the formation of invasive fronts were detected. While the Panc1 cell populations already showed 20 invasive fronts at day 1, with no clear differences between the parental, Holo-, and Paraclone cells, the Panc89 cell variants only exhibited five invasive fronts on day 5, which did not change until day 7 (Figure 4E). Interestingly, the number of invasion fronts declined in the Panc1 cells, being most pronounced in the Panc1 Holoclone cells. However, invasion distances concomitantly increased in the Panc1 cells, rising from 260 to 350 µm on day 1, to about 400 to 500 µm on day 3. In line with the results of the invasion fronts, the most pronounced increase in the invasion distance from day 0 to day 3 was observed in the Panc1 Holoclone cells. In contrast, the invasion distances of all the Panc89 cell variants were much shorter (about 100 µm) and did not change over time (Figure 4F). 

Despite the significantly higher cell migration of the Panc89 Holoclone cells compared to the Panc89 Paraclone cells, no clear differences regarding cell invasion were observed between the respective Holo- and Paraclone cell populations, which also applied to the Panc1 cell variants. Overall, these data underscore that the EMT rather than the CSC phenotype predominantly determines the migration and invasion mode of PDAC cells. Thus, these data underscore that the mesenchymal-like Panc1 cells use a fast mesenchymal mode of single-cell invasion, while the Panc89 cells exhibiting an epithelial phenotype use a slow invasion mode in clusters.

As PDAC metastasizes, predominantly in liver, lungs, and peritoneum [4,5,6], a further analysis was conducted to determine whether the different PDAC cell variants, and particularly the CSC and non-CSC populations, differ with respect to their adhesion properties to liver and lung endothelial cells as well as mesothelial cells. However, no clear differences between Holoclone and Paraclone cells of either PDAC cell line were observed. In general, the Panc89 cell variants showed a more pronounced cell adhesion to lung endothelial HuLEC-5a cells (Appendix A). Overall, these findings indicate that the CSC phenotype of PDAC cells rather marginally impact cell adhesion to organ-specific endothelial cells.

### 3.2. Tumorigenicity and Metastasis Analysis of Panc1 and Panc89 Holo- or Paraclone Cells In Vivo

#### 3.2.1. Panc1 and Panc89 Cell Variants Essentially Differ with Respect to Their Metastatic Capacity In Vivo

Finally, to assess whether CSC–EMT phenotypes and the associated functional differences identified in vitro impact their tumorigenic and metastatic behavior, a tumorigenicity and metastasis assay with intrasplenic inoculation with low numbers of either Panc1 and Panc89 Holo- or Paraclone cells (1 × 10^4^ cells/each) was performed. 

As shown in Figure 5A, animals inoculated with the Panc89 Holo- or Paraclone cells survived at a significantly shorter rate than animals inoculated with the Panc1 Holo- or Paraclone cells. The median survival rate of mice inoculated with the Panc89 Holo- and Paraclone cells was 74 days and 66 days, respectively. In contrast, animals inoculated either with the Panc1 Holo- or Paraclone cells showed a median survival rate of 146 days, twice the length of the Panc89 Holo- and Paraclone cell-inoculated animals (Figure 5A). Of note, animals inoculated with Panc1 cell variants were not sacrificed because of reduced health status due to high tumor burden; rather, for these mice, a slow tumor progression, which had almost not changed during the entire observation time, was noticed. An examination of the macroscopic tumor manifestation revealed clear differences between the Panc1 and Panc89 cell variants, as well as between the Holo- and Paraclone cells. An ultrasound examination revealed macroscopic tumor formation in five out of ten animals with a median tumor area of 11.68 mm^3^ for the Panc1 Holoclone tumors, and only in two out of ten animals for the Panc1 Paraclones tumors (with no median macroscopic tumor area detectable because one of two tumors could not be measured via ultrasound). In contrast, in nine out of ten animals inoculated with the Panc89 Holoclone cells, tumors could be detected with a median tumor area of 137.9 mm^3^. The inoculation with the Panc89 Paraclone cells led to slightly less tumor formation, as only tumors with a median macroscopic tumor area of 122.1 mm^3^ were detected in seven out of ten animals (Figure 5B,C). Tumor sizes of the Panc89 Holoclone and Paraclone tumors were significantly larger compared to the respective cell variants of the Panc1 cells (*p* = 0.01). Thus, shorter survival times of animals inoculated with the Panc89 cell variants were associated with higher tumor burden compared to the Panc1 Holo- and Paraclone cells.

Clear differences could also be noted regarding organ manifestation. While Panc1 Holo- and Paraclone tumors were predominantly found in the pancreas, and less frequently also in the liver and lungs, the Panc89 Holo- and Paraclone tumors were predominantly detectable in the peritoneum, followed by the pancreas, liver, and lungs (Figure 5C). In detail, inoculation with Panc1 Holoclone cells clearly led to more tumors compared to Panc1 Paraclone tumors, which were predominantly formed in the pancreas (five out of ten animals) and liver (three out of ten animals), while the inoculation with Panc1 Paraclone cells only led to tumors in the pancreas and lungs of one out of ten animals (Figure 5C). Macroscopically, the inoculation with the Panc89 Holoclone cells also led to a higher number of tumors compared to the Panc89 Paraclone cells, and also compared to all Panc1 cell variants. The Panc89 Holoclone tumors were found in the peritoneum of eight out of ten animals, in the pancreas of two out of ten animals, and in the liver and lungs of one out of ten animals. The Panc89 Paraclone tumors were found in the peritoneum of four out of ten animals, in the pancreas of two out of ten animals, and in the liver of one out of ten animals (Figure 5C).

Having determined clear differences regarding the frequency and site of metastases of the inoculated animals, the pancreatic, liver, lung, and peritoneal tissues were stained for human PanCK (Appendix A) to assess the microscopical metastatic burden in terms of size and number of formed microscopic tumors. 

In line with the size of the macroscopic tumors, the median tumor areas of both the Panc89 Holo- and Paraclone tumors were larger than the Panc1 Holo- and Paraclone tumors (Figure 5D). The inoculation with Panc89 Paraclone cells led to the largest microscopic tumor areas with a median of about 25 mm^2^, followed by the Panc89 Holoclone tumors with a median tumor area of about 8 mm^2^. The Panc1 Holo- and Paraclone cells formed tumors with a median tumor area of less than 5 mm^2^ (Figure 5D). In addition, it could be noted that five out of eight Panc89 Paraclone tumors were characterized by a pronounced cyst formation (Appendix A).

Of note, the inoculation with Panc1 Holoclone cells led to the highest number of microscopic lesions, with a total of 31 microscopic lesions in nine out of ten animals, while only nine microscopic tumors were found in eight out of ten animals inoculated with Panc1 Paraclone cells. The highest number of microscopic Panc1 Holoclone tumors were found in the pancreas (22/31), while Panc1 Paraclone lesions were mostly found at other sites (brown adipose tissue, duodenum, undefinable in five of nine metastatic lesions) (Figure 5E). With 17 microscopic tumors in ten out of ten mice, the number of microscopic Panc89 Holoclone tumors was much lower compared to Panc1 Holoclones, and organ manifestation was also very different as most tumors were found in the peritoneum (14/17 metastatic lesions), like the macroscopic tumors (Figure 5E). In addition, the inoculation with Panc89 Holoclone cells led to a twofold higher formation of microscopic tumors (17 tumors) compared to the inoculation with Panc89 Paraclone cells (eight tumors), which was similar to Panc1 Paraclone cells (Figure 5E). The inoculation with Panc89 Paraclone cells led to a slightly higher prevalence of tumor formation in the peritoneum compared to other sites (three out of eight lesions).

Overall, the inoculation with Panc89 Holo- and Paraclone cells led to a more pronounced metastatic burden in terms of larger metastases, which predominantly manifested in the peritoneum, compared to Panc1 Holo- and Paraclone cells, which led to smaller tumors mostly in the pancreas. A higher tumor burden after the inoculation with Panc89 Holo- and Paraclone cells was associated with significantly shorter survival times and is clearly in line with the more pronounced proliferative activity of the cells. However, the inoculation with Panc1 Holoclone cells led to the highest total number of microscopic tumoral lesions, which is in line with the more pronounced invasive phenotype and slower proliferative activity of the cells. The fact that the Panc89 Paraclone cells also led to a pronounced tumor burden, which was not observed for the two Panc1 cell variants, underscores a higher plasticity of these cells. Altogether, these findings indicate clear differences in the metastatic capacity between mesenchymal-like and epithelial/hybrid CSC populations.

#### 3.2.2. Panc1 and Panc89 Holo- and Paraclone Tumors Exhibit Differences in EMT and CSC Marker Expression

To verify the in vitro-identified CSC–EMT phenotypes of the Panc1 and Panc89 cell variants in vivo, IHC stainings of resected tissues was performed to analyze protein levels of the CSC markers Nestin and SOX2, as well as the EMT markers E-cadherin, L1CAM, ZEB1, and ZEB2. 

First, the PanCK staining of tumors revealed different morphologies of the Panc1 and Panc89 Holo- and Paraclone tumors. While the Panc1 Holo- and Paraclone tumors were characterized by a scattered arrangement of single tumor cells, the Panc89 Holo- and Paraclone tumors showed positively stained cell clusters. The Panc89 Holoclone tumors especially presented distinct PanCK positive cell clusters surrounded by the tumor stroma. The cell clusters in the Panc89 Paraclone tumors were larger, with less intense PanCK staining, but huge stromal areas were observed (Appendix A).

In confirmation with the gene expression data, the Panc1 Holoclone tumors showed a negative to low expression of E-cadherin (Figure 6A). However, the Panc1 Paraclone tumors showed slightly but significantly higher E-cadherin expression levels, with a frequency and intensity score of 2 compared to Panc1 Holoclone tumor tissues (Figure 6A). In the Panc89 Holo- and Paraclone tumors, E-cadherin was highly expressed at comparable levels, confirming the epithelial phenotype of the tumors (frequency and intensity score of about 4) (Figure 6A).

In line with the in vitro data, the L1CAM expression in the Panc1 Holoclone tumors was widely detectable (frequency score 2), while the Panc1 Paraclone tumors showed significantly lower L1CAM levels (frequency and intensity score 1, Figure 6B) compared to Panc1 Holoclone tumors. The Panc89 Holoclone tumors showed a very faint expression of L1CAM compared to the Panc89 Paraclone tumors, exhibiting the highest L1CAM expression level of all tumors analyzed (frequency and intensity score of 3 and 2) (Figure 6B), confirming data from the in vitro analysis. However, L1CAM expression could also be noted in some Panc89 Holoclone tumors (Appendix A).

An IHC analysis of ZEB1 revealed that the Panc1 Holoclone tumors exhibited strong ZEB1 levels (Frequency and Intensity score 4), while the Panc1 Paraclone tumors were characterized by a low to negative ZEB1 (Figure 6C), which was not in line with the *ZEB1* gene expression in vitro, demonstrating similar RNA levels of *ZEB1* in both Panc1 cell variants. In both Panc89 Holo- and Paraclone tumors, a high ZEB1 expression with a frequency and intensity score of 3-4 was noted (Figure 6C), which also differed from the absent *ZEB1* gene expression in vitro. Of note, ZEB1 and ZEB2 levels were mostly located in the nuclei, which indicates their functional activity, and both markers could not only be detected in PDAC cells but also in stroma cells. The Panc1 Holoclone tumors showed a significantly higher level of ZEB2 (frequency score of 2–3 and intensity score of 1) compared to the Panc1 Paraclone tumors (frequency and intensity score of 1, Figure 6D). The Panc89 Holoclone tumors showed a slightly lower ZEB2 expression (frequency and intensity score of 3 and 2, respectively) compared to the Panc89 Paraclone tumors, with a frequency and intensity score of 4 and 3, respectively (Figure 6D).

Finally, the expression of the CSC markers Nestin and SOX2 was assessed in all tumors. In the Panc1 Holoclone tumors, the Nestin level was high (frequency score 2–3 and intensity score 4), significantly differing from the Panc1 Paraclone tumors, which showed no or only a weak Nestin expression (frequency and intensity score 1), reflecting the different expression patterns identified in vitro. Additionally, in line with the in vitro findings, both Panc89 Holo- and Paraclone tumors exhibited no or only very faint Nestin levels (Figure 6E).

However, differing from the in vitro findings, SOX2 expression could be detected in the Panc1 Holoclone tumors (frequency and intensity score of 2 and 3, respectively) and was higher compared to the Panc1 Paraclone tumors (frequency and intensity score of 1 and 2, respectively, Figure 6F). Again, in line with the in vitro data, the SOX2 expression in the Panc89 Holoclone tumors was significantly higher (frequency and intensity score of 2 and 2, respectively) compared to the Panc89 Paraclone tumors (frequency and intensity score of 1 and 1, respectively, Figure 6F). 

Of note, four out of seventeen Panc89 Holoclone tumors showed L1CAM positive areas. Correlating the L1CAM and SOX2 levels in all Panc89 Holoclone tumors, no significant correlation could be noted. However, by analyzing Panc89 Holoclone tumors expressing L1CAM (frequency score of 2–3) regarding their SOX2 status, the L1CAM expression in these tumors was significantly associated with low/absent SOX2 expression levels (Appendix A), which is indicative of the phenotype of Panc89 Paraclone cells.

In summary, the IHC analysis widely confirmed the gene expression profile of the cells identified in vitro, confirming the mesenchymal-like Panc1 Holoclone cells with a Nestin-dominated CSC phenotype, while the mesenchymal-like Panc1 Paraclone cells showed less/no stemness characteristics. Contrary, the Panc89 Holoclone cells mostly exhibited an epithelial SOX2-dominated CSC phenotype, whereas the Panc89 Paraclone cells exhibited a partial/hybrid EMT cell phenotype with less CSC marker expression. However, certain variations from the CSC–EMT marker expression patterns identified in vitro could be noted as well, e.g., differing expression of L1CAM, SOX2, ZEB1, and ZEB2 in the Panc89 Holo- and Paraclone tumors and SOX2 expression in Panc1 Holoclone tumors, indicating the phenotypic switching of CSCs and non-CSCs upon the influence of the distinct tumor microenvironments.

Overall, these data indicate that different CSC–EMT phenotypes of PDAC cell populations might yield different metastatic manifestations, which are associated with different survival times. 

## 4. Discussion

Cancer cell plasticity and tumor heterogeneity, e.g., manifested by the presence and interplay of CSC and non-CSC populations, along with the pronounced tumor stroma, are associated with a poor prognosis and increased resistance to chemotherapy for PDAC [62,63,64,65,66,67,68,69,70,71,72,73,74]. Besides, CSCs and EMT are both linked to cancer progression and early metastasis, which also applies to PDAC [18,19,20,75]. CSCs are characterized by the unique ability of self-renewal, the initiation of tumoral lesions at primary and secondary sites, the capability to resist cell death induction, and to give rise to plastic, highly proliferative transit-amplifying progenitor cell populations [21,22,23,76,77,78,79,80]. As they also generate more or less differentiated cancer cells, CSCs are one important origin of tumor cell heterogeneity [20,24,25,26,27,81]. The link between EMT and MET processes and CSCs has been already shown [18,19,20,75]. Yet, it is still poorly understood whether CSC and EMT phenotypes are always interconnected, how CSC phenotypes of epithelial and mesenchymal-like tumor cells are characterized, and whether these might be related to different functional malignant outcomes.

To gain a better understanding of plasticity in PDAC with a particular focus on CSCs and EMT and its contribution to malignancy-associated properties, two different PDAC cell models were comparatively analyzed. Mesenchymal-like Panc1 cells derived from a primary PDAC have presumably already undergone EMT, while the Panc89 cells originate from a lymph node metastasis and have most likely undergone EMT and MET leading to metastatic outgrowth in the lymph node [82]. Isolated Panc1 and Panc89 Holo- and Paraclone cells, as well as the two related parental cell populations, were comprehensively analyzed by expression and functional analyses in vitro and in vivo. First, CFAs and the expression analysis of CSC markers revealed different CSC phenotypes in the PDAC cell populations. While the Panc1 Holoclone cells were characterized by high Nestin and absent SOX2 expression, the Panc89 Holoclone cells exhibited an inverse expression pattern, showing the SOX2-dominated phenotype. Moreover, the CSC phenotype of Panc89 Holoclones was associated with a higher colony-formation ability compared to Panc1 Holoclones, indicating a more pronounced self-renewal capacity. Of note, pathways involved in the activity of cell cycle checkpoints, the transcriptional regulation of TP53, and Mueller Plurinet, seemed to generally play a role in CSC populations of PDAC, as these genes were more highly expressed in both Panc1 and Panc89 Holoclone cells compared to related Paraclones. Mueller Plurinet is a protein–protein network shared by pluripotent cells and has become a prominent classifying system for pluripotency and self-renewal properties [59]. Overall, these data support the existence of distinct CSC phenotypes in PDAC cells.

Furthermore, clear differences with respect to the EMT phenotype were noted. The Panc89 cell variants were predominantly characterized by an epithelial phenotype, exemplified by a high E-cadherin expression. In contrast, all Panc1 cell variants were characterized by a low E-cadherin expression and a marked expression of mesenchymal markers, underscoring the mesenchymal-like phenotype. As the Panc89 Paraclone cells showed a high E-cadherin expression, low Vimentin expression, but considerable expression of L1CAM, ZEB1, and ZEB2, which was higher compared to the other Panc89 cell variants, but still lower compared to the Panc1 cell variants (despite ZEB2 expression), these cells seemed to exhibit a partial/hybrid EMT phenotype, as described in other tumor cell models [83,84,85,86]. Moreover, these phenotypes were widely confirmed using IHC staining of the tumors formed in vivo by the inoculated Holo- and Paraclone cell populations. However, notable plasticity-based changes could be observed, particularly for the CSC populations. The Panc89 Holoclone tumors showed areas with a high L1CAM and low SOX2 expression, but also showed areas with ZEB1 or ZEB2 expression. Furthermore, the Panc1 Holoclone tumors showed an elevated SOX2 expression level, which was absent in vitro. These phenotypic switches of CSC populations, but also non-CSC populations (particularly Panc89 Paraclone tumors), and thus differing phenotypes from the in vitro analysis, could be explained by the exposure of the tumor cells to different microenvironmental niches. Thus, myofibroblasts and macrophages, both important stroma cell populations in pancreatic tumors and metastases [87,88], were already shown to induce L1CAM expression in PDAC cells [89,90,91] Altogether, these data indicate a robust phenotype stability of Holo- and Paraclones of either PDAC cell line under constant environmental conditions. Although, when the latter are changing, this might lead to a phenotypic switching of the tumor cells, presumably forced by the altered environmental factors the tumor cells are exposed to, as observed in the in vivo tumors.

Even though the transcriptomic profiling revealed that both Panc1 and Panc89 Holoclone cells show stemness associated Mueller Plurinet activity, we could identify two distinct EMT–CSC phenotypes: a mesenchymal-like Nestin-dominated (Panc1 Holoclone cells) and an epithelial SOX2-dominated CSC phenotype (Panc89 Holoclone cells). Of note, both phenotypes were associated with clearly different functional properties. In addition to the more pronounced colony formation, the Panc89 Holoclone cells showed a clearly faster growth behavior compared to the Panc89 Paraclone cells, but also to the Panc1 Holoclone cells, which is in line with the fact that the proliferative activity of epithelial cells is generally higher than those of mesenchymal cells [14,92]. In line with a higher cell turnover of epithelial Panc89 cell variants, parental Panc89 cells, as well as their Holoclone and Paraclone cells, showed a better response towards Gemcitabine compared to mesenchymal-like Panc1 cell variants. These differences regarding the response to chemotherapeutic drugs were in accordance with those having already assigned to the classical and basal-like phenotype of PDAC cells [60,93]. Of note, mesenchymal-like CSC populations (Panc1 Holoclone cells) showed an even more resistant phenotype compared to their parental cells and non-CSCs, while epithelial CSCs (Panc89 Holoclone cells) showed the strongest treatment response of all cell variants. These data support the rationale for using chemotherapy as an effective treatment for fast proliferating tumor cells, which, however, fails to eliminate tumor cells with slow proliferative activity [94,95,96]. Furthermore, these findings underscore the heterogeneity of CSCs, which manifest in different treatment sensitivities.

Showing a slower cell growth, Panc1 cells exhibited a faster cell invasion in a typical mesenchymal-like cell manner with increased numbers of invasive fronts and longer invasion distances, underscored by the activity of EMT and invasion-associated genes, shown in the transcriptomic analysis. In contrast, the Panc89 Holoclone cells used a high cell-cell contact cluster mode of slow cell invasion, again fitting together with the epithelial phenotype of these cells [14,82,92]. Despite the more pronounced invasion ability, the Panc1 cell variants were characterized by slower cell migration compared to the Panc89 cell variants. This observation might be explained by the fact that the Panc1 cell variants exhibited a higher expression of proteases such as matrix metalloproteinase (MMP)-2 and MMP-9 compared to Panc89 cell variants (Appendix A). Both MMPs have the ability to degrade collagen, which is a major structural component of basement membranes, and Matrigel, which is used in the invasion assays, and thus facilitate cellular invasion [97,98].

Overall, the more pronounced invasive capacity of the Panc1 Holoclone cells, along with the elevated expression of mesenchymal proteins like Vimentin [99,100,101,102,103,104] and L1CAM [89,105,106,107], which have been associated with an increased invasive potential in a variety of cancers like PDAC, are clearly in line with the highest number of microscopic tumoral lesions detected in vivo. Both Panc1 cell variants showed a comparable expression pattern of epithelial and mesenchymal markers, as well as a similar invasion velocity, but the Panc1 Holoclone cells led to the formation of more and larger tumors compared to Panc1 Paraclones; this underscores the fact that the presence of CSC-like properties in mesenchymal-like PDAC cells is a prerequisite for metastatic tumor formation.

Although the Panc1 Holoclone cells also led to a higher number of microscopic lesions compared to the Panc89 Holoclone cells, the inoculation with Panc89 Holoclone cells yielded the most pronounced overall tumor burden with the highest number of macroscopic tumors of large sizes compared to all other cell variants, which is clearly in line with the highest proliferative and self-renewal ability of these cells observed in vitro. 

The fact that the inoculation with Panc1 and Panc89 Paraclone cells led to tumor formation, albeit to a lesser extent, supports the view of a high plasticity in these cell populations, implying that non-CSCs can gain CSC properties. Notably, despite this possible gain of CSC properties in the Panc1 and Panc89 Paraclone cells, the overall expression of CSC markers was rather low in these tumors. Thus, further studies are needed to investigate the role of other CSC markers and factors, e.g., those provided by the tumor microenvironment (see above), that determine the phenotypic switching of non-CSCs and CSCs. 

In contrast to Panc1 tumors, which predominantly occurred in the pancreas, tumors formed by the Panc89 cell variants predominantly manifested in the peritoneum. Previous studies have demonstrated that PDAC patients with peritoneal recurrences exhibit significantly shorter disease-free survival times and worse overall prognoses compared to PDAC patients with, e.g., pulmonary recurrences [7], which is consistent with the shorter survival times of animals that were inoculated with Panc89 cell variants and formed peritoneal metastasis.

Since the adhesion assays did not reveal any conclusive differences regarding PDAC cell adhesion to organ-specific endothelial cells between Holo- and Paraclone populations of either cell line, other factors seemed to be more crucial in determining the tumor manifestation patterns. The fact that mesenchymal-like Panc1 cells are derived from the primary tumor, together with the fac that their gene activity is associated with EMT and invasion, suggests that these cells have undergone EMT to leave the primary tumor. However, whether these cells would ever have been able to form metastases in this patient remains unsolved. Furthermore, it may explain why most Panc1 Holoclone tumors were found in the pancreas, indicating that these cells are still optimally adapted to their original tissue.

In contrast, the epithelial Panc89 cells originate from a lymph node metastasis; thus, these cells have proven their disseminating potential to leave the primary tumor (either after EMT using a mesenchymal-like invasion mode or while maintaining an epithelial/hybrid cell stage using a cluster-like mode) and to metastasize at a secondary site (e.g., after MET). Thus, they have managed to survive all steps of the metastatic cascade, including adaptation to novel microenvironments. It can be speculated whether the patient, from which the Panc89 cells were derived, had developed peritoneal metastases during the course of the disease, which would be consistent with tumor manifestation in our in vivo model. Furthermore, it would be of great interest to investigate whether primary tumors of PDAC patients who developed peritoneal metastases contain more epithelial CSCs, and whether primary tumors of patients with other metastatic sites show a more mesenchymal-like CSC phenotype.

As outlined before, both Nestin and SOX2 have been associated with CSC phenotypes and linked to malignancy-associated properties. Thus, in a murine PDAC model, the shRNA-mediated reduction of Nestin expression led to decreased tumor volume and hepatic metastases [37,38], which is in line with our results demonstrating that Nestin-expressing Panc1 Holoclone cells showed a higher self-renewal capacity and invasive properties in vitro, and formed a higher number of tumors in vivo compared to Panc1 Paraclone cells. However, the Panc89 Holoclone cells lacked Nestin expression but were characterized by a high SOX2 expression level compared to the Panc89 Paraclone cells, as well as compared to all Panc1 cell variants, which is associated with the highest proliferative and self-renewal ability of all cell variants tested, underscoring the diversity of CSC phenotypes. Of note, the Panc1 and Panc89 Holoclone tumors exhibited similar amounts of SOX2, which supports our understanding of the role of the microenvironment as a determining factor of CSC properties [20,24,25,26,27,108]. Thus, it can be speculated that the elevation of SOX2 might also be involved in the outgrowth of Panc1 Holoclone tumors [32,109,110].

Overall, these findings support the view that the CSC phenotype does not seem to determine homing to secondary sites rather than outgrowth and the self-renewal of the tumor cells, while the EMT phenotype seems to determine the dissemination mode. The metastatic propensity of PDAC cells seems, therefore, to be the result of a fine-tuned interplay of both phenotypes.

## 5. Conclusions

In summary, these results support the view that mesenchymal-like CSC have a strong propensity to colonize secondary sites and to form a higher number of (small) tumoral lesions, which, however, does not ultimately lead to a fast disease progression associated with a short survival rate. In contrast, the epithelial CSC phenotype seemed to be associated with a slow cell invasion but the concomitant advantage of rapid tumor outgrowth, resulting in a fast increase in life-threatening tumor burden (exemplified by the highest number of macroscopic tumors and shorter survival). Overall, our data support the view that different CSC phenotypes exist in PDAC, which are associated with distinct EMT phenotypes of PDAC cells, and essentially determine PDAC cell fate and function, as well as treatment responses. However, because of the fact these findings have been obtained only with two PDAC cell lines and their related CSC and non-CSC variants, studies with Holo- and Paraclone cell variants derived from other PDAC cell lines have to be conducted to corroborate these results.

## Figures and Tables

**Figure 1 cancers-16-00686-f001:**
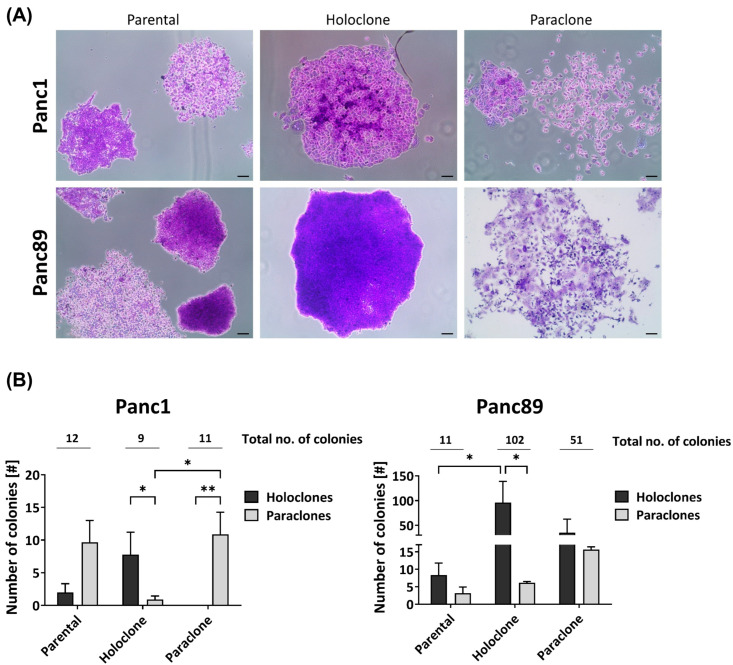
Panc1 and Panc89 cell variants exhibit differences in their colony-formation ability. Parental Panc1 and Panc89 as well as their respective Holo- and Paraclone cell variants were analyzed regarding their colony-formation ability via CFA. 400 cells of each cell line were seeded in 6-well plates and after eight to eleven days, colonies were fixed with paraformaldehyde and stained with crystal violet. The colonies formed were morphologically characterized regarding Holo- and Paraclones. (**A**) Typical colony morphologies generated by parental cells as well as respective Holo- and Paraclones. (**B**) Number of Holo- and Paraclones per well formed by parental Panc1, Holo-, and Paraclone cell variants (left), as well as parental Panc89, Holo-, and Paraclone cell variants (right). Y axis in B (right) is segmented with 0–18 for the bottom part and 30–200 for the upper part. Data are presented as mean with SEM from *n* = 3 independent experiments. Significances are indicated by asterisks: *p* ≤ 0.033 = *, *p* ≤ 0.002 = **. Scale bar = 100 µm (no. = number; SEM = standard error of means).

**Figure 2 cancers-16-00686-f002:**
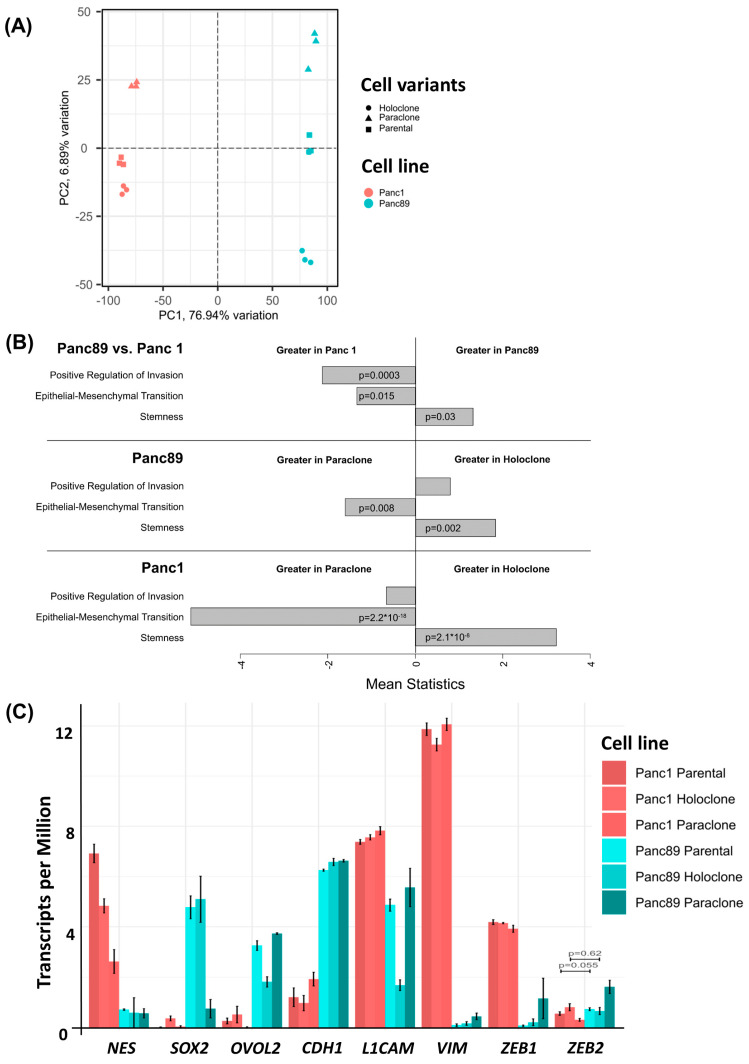
Panc1 and Panc89 cells as well as their derived Holo- and Paraclone cells show distinct CSC and EMT marker signatures. (**A**) Principal Component Analysis (PCA) based on the 10% most varying transcripts across all of parental Panc1 and Panc89, Holo- and Paraclone cell samples using the R package PCAtools (v2.10.0). PCA was performed for the highest variation in the first principal component (PC1) on the X axis, and for the lower variations in the second principal component (PC2) on the Y axis. All cell variants and cell lines are denoted by color (red = Panc1 cell variants, blue = Panc89 cell variants) and symbol, respectively. (**B**) Gene Set Variation Analysis (GSVA) of gene sets associated with invasion, EMT, and stemness in parental Panc1 and Panc89 cells as well as their derived Holo- and Paraclone cells. (**C**) Transcriptomic analysis of CSC- and EMT-associated marker genes in parental Panc1 and Panc89 as well as Holo- and Paraclone cells. *CDH1* (epithelial), *VIM*, and *L1CAM* (mesenchymal) represent EMT marker genes; *ZEB1* and *ZEB2* represent marker genes for EMT induction; and *NES*, *OVOL2*, and *SOX2* represent CSC marker genes. Data are represented as means with SD. Gene expression analysis by qPCR was performed for (**D**) *NES*, *SOX2* and *OVOL2*, (**E**) *CDH1*, *L1CAM* and *VIM* as well as (**F**) *ZEB1* and *ZEB1*. Parametric data are shown as mean with SD; non-parametric data are presented as median with interquartile range. Significances in (**D**–**F**) are indicated by asterisks: *p* ≤ 0.002 = **, *p* ≤ 0.001 = ***. (**G**) Immunofluorescence staining of EMT and CSC markers was performed for Panc1 cell variants with the CSC marker NESTIN and the EMT inducer ZEB2, while the CSC marker SOX2 and the mesenchymal EMT marker L1CAM were stained in Panc89 cell variants. Cell nuclei were stained with Hoechst 33342. Representative images from *n* = 3 independent experiments are shown (scale bar = 200 µm). Every analysis was performed with *n* = 3 independent experiments. (Holo = Holoclone cells, Para = Paraclone cells, SD = standard deviation).

**Figure 3 cancers-16-00686-f003:**
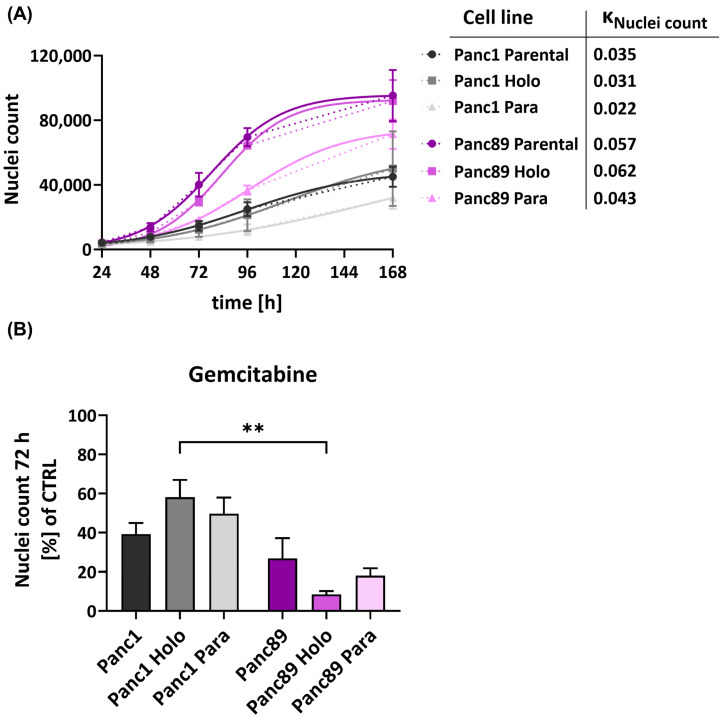
Panc1 and Panc89 cell variants show different behavior regarding cell growth and response to chemotherapy. (**A**) For growth rate analysis of parental Panc1 and Panc89 cells as well as their derived Holo- and Paraclone cells, 5 × 10^3^ cells of each cell variant were seeded in a 96-well plate in triplicates and nuclei number was monitored via Hoechst 33342 staining for 168 h. Data are depicted as total number of counted cell nuclei (mean with SEM). k_Nuclei count_ = (logistic) growth rate. (**B**) Panc1 and Panc89 cell variants were seeded at 5 × 10^4^ cells in duplicates in 96-well plates. After 24 h, cells were left untreated or treated with 0.0038 µM Gemcitabine. After 72 h, cells were stained with Hoechst 33342 for nuclei count analysis and nuclei number of treated cells was normalized to nuclei number of untreated cells. Treatment data are shown as mean with SEM. Significances are indicated by asterisks: *p* ≤ 0.002 = **. Every analysis was performed with independent experiments *n* = 3. (CTRL = untreated control, SEM = standard error of means).

**Figure 4 cancers-16-00686-f004:**
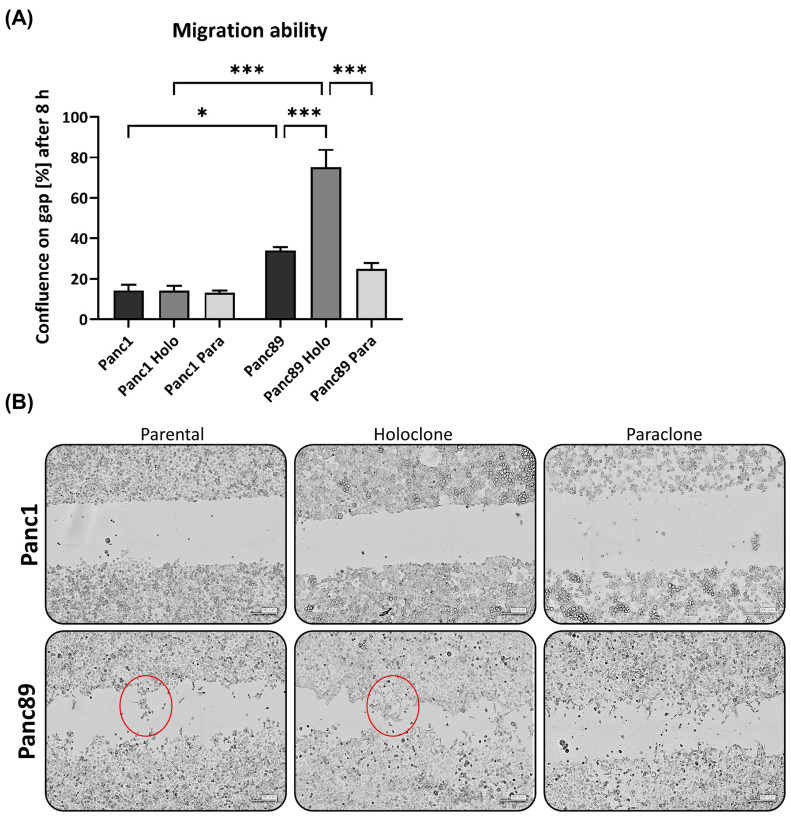
Panc1 and Panc89 cell variants clearly differ in their migration and invasion abilities. (**A**) Migration ability of parental Panc1 and Panc89 as well as Holo- and Paraclone cells was determined by monitoring the increase of cell confluence on a cell-free gap for 8 h. (**B**) Representative images of cell migration of Panc1 and Panc89 cell variants after 8 h. (**C**) Representative images of spheroid formation and invasion ability of Panc1 and Panc89 cell variants. Examples for (**C**) migration and (**D**) invasion patterns are highlighted with red circles. (**D**) Spheroid size for Panc1 cell variants on day 0 and day 3 and Panc89 cell variants on day 0 and day 7. (**E**) Analysis of the number of invasive fronts of Panc1 cell variants (day 1, day 3) and Panc89 cell variants (day 5, day 7). (**F**) Analysis of invasion distance of Panc1 cell variants (day 1, day 3) and Panc89 cell variants (day 5, day 7). Data are presented as mean with SEM from *n* = 3 independent experiments. Scale bar (**B**) 200 µm; scale bar (**C**) Panc1 cell variants 200 µm, Panc89 cell variants 100 µm. Significances are indicated by asterisks: *p* ≤ 0.033 = *, *p* ≤ 0.002 = **, *p* ≤ 0.001 = *** (Holo = Holoclone cells, Para = Paraclone cells, SEM = standard error of means).

**Figure 5 cancers-16-00686-f005:**
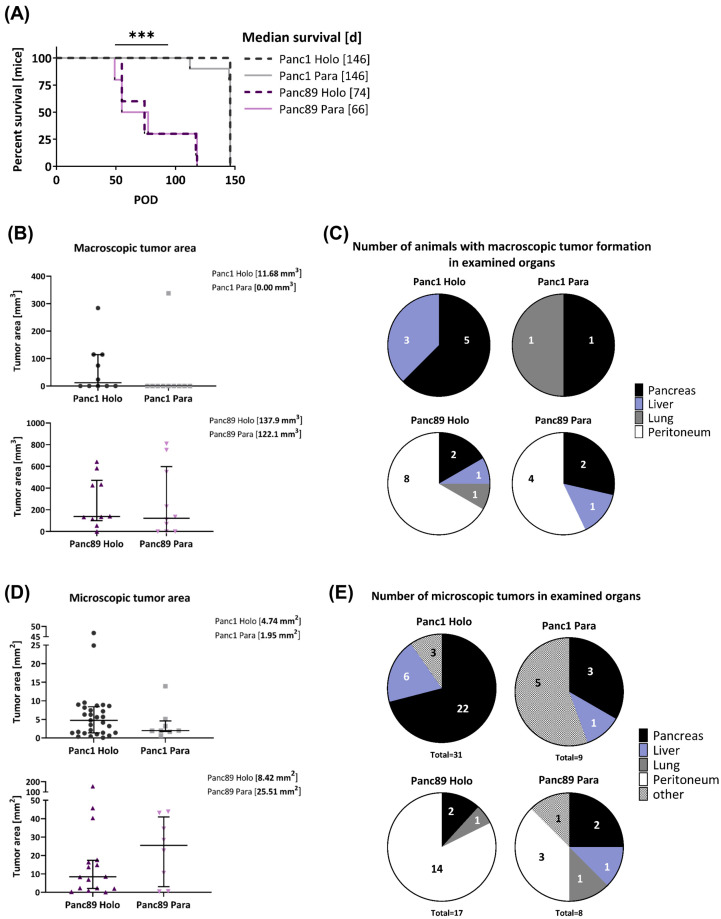
Panc1 and Panc89 cell variants differ with respect to their metastatic capacity in vivo. Mice were inoculated intrasplenically with 1×10^4^ Panc1 or Panc89 Holo- or Paraclone cells (group of ten mice per cell line). (**A**) Kaplan–Meier–Survival analysis of animals inoculated with Panc1 or Panc89 Holo- and Paraclone cells. Significance is indicated by asterisks: *p* ≤ 0.001 = ***. (**B**,**C**) Macroscopic tumor areas [mm^3^] and organs of macroscopic tumor manifestation in Panc1 and Panc89 Holo- or Paraclone cell-inoculated mice. (**D**,**E**) Microscopic tumor areas [mm^2^] as well as number and location of microscopic tumors after PanCK staining of tissue sections. Data for macroscopic (**B**) and microscopic (**D**) tumor areas are shown as median with interquartile range. (Holo = Holoclone cells, Para = Paraclone cells, POD = post-operative day).

**Figure 6 cancers-16-00686-f006:**
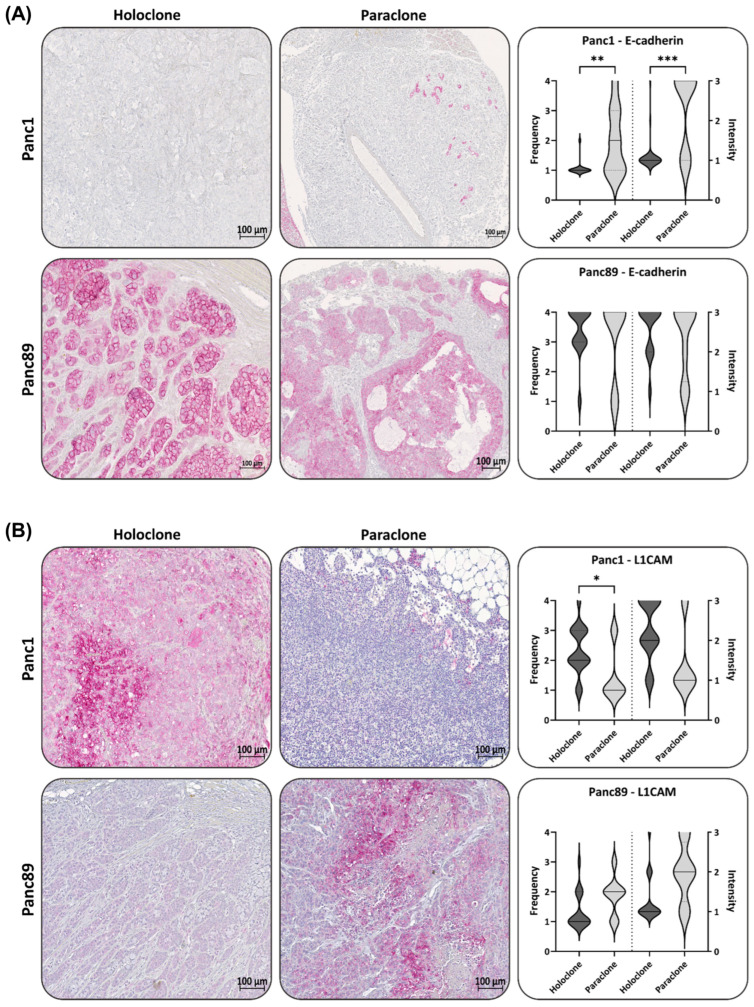
Panc1 and Panc89 Holo- and Paraclone tumors exhibit differences in EMT and CSC marker expression. Mice were inoculated intrasplenically with 1 × 10^4^ Panc1 or Panc89 Holo- or Paraclone cells (group of ten mice per cell line) and resected tissues and tumor lesions were stained for the epithelial marker (**A**) E-cadherin, the mesenchymal marker (**B**) L1CAM, the EMT inducers (**C**) ZEB1 and (**D**) ZEB2, and the CSC markers (**E**) Nestin and (**F**) SOX2. Representative pictures of stained tissues of Panc1 and Panc89 Holo- or Paraclone tumors are presented in the left and middle picture panels; the right panel shows the statistical analysis (median with range) of frequency and intensity score of positively stained cells. The upper row shows the data of analyzed Panc1 Holo- and Paraclone lesions; the lower panel shows the data of Panc89 Holo- and Paraclone lesions. Significances are indicated by asterisks: *p* ≤ 0.033 = *, *p* ≤ 0.002 = **, *p* ≤ 0.001 = ***. Scale bar = 100 µm.

**Table 1 cancers-16-00686-t001:** Target genes, primer sequences, and annealing temperatures used for quantitative polymerase chain reaction.

Target	5′-3′ Sequence	Annealing [C°]
*CDH1* (E-cadherin) **	fw-TGCTCTTGCTGTTTCTTCGG rv-TGCCCCATTCGTTCAAGTAG	55
*GAPDH* *	fw-TCCATGACAACTTTGGTATCGTGG rv-GACGCCTGCTTCACCACCTTCT	58
*L1CAM* **	fw-GAACTGGATGTGGTGGAGAG rv-GAGGGTGGTAGAGGTCTGGT	58
*NES* (Nestin) *	fw-GAAACAGCCATAGAGGGCAAA rv-TGGTTTTCCAGAGTCTTCAGTGA	58
*OVOL2* (ZNF339) **	fw-GGGACAAGCTCTACGTCTGC rv-GTCTGTCCTCCCCTTCCTTC	58
*SOX2* *	fw-TCCCATCACCCACAGCAAATGA rv-TTTCTTGTCGGCATCGCGGTTT	58
*VIM* (Vimentin) **	fw-TCCAAGTTTGCTGACCTCTC rv-TCAACGGCAAAGTTCTCTTC	58
*ZEB1* *	fw-TCCATGCTTAAGAGCGCTAGCT rv-ACCGTAGTTGAGTAGGTGTATGCCA	61
*ZEB2* **	fw-CACATCAGCAGCAAGAAATG rv-AAACCCGTGTGTAGCCATAA	58

* = Primers purchased from Eurofins Genomics GmbH (Ebersberg, DE). Stocks of forward (fw) and reverse (rv) primers diluted at 1 pmol/µL in nuclease-free ddH_2_O. ** = Primers purchased from RealTime Primers (via Biomol, Hamburg, Germany), provided as stocks of 50 µM primer mix (containing fw and rv primer) in nuclease-free ddH_2_O supplemented with 10 mM Tris-HCL and 0.1 mM EDTA (pH 7.5).

**Table 2 cancers-16-00686-t002:** List of antigen retrieval conditions.

Antigen	Buffer (pH)	Temperature, Time
anti-E-cadherin	S1699 (pH 6.1)	Steaming 121 °C, 10 min
anti-L1CAM	EDTA (pH 8.0)	1. microwave broiling 800 watts 2. microwave 560 watts, 3 × 5 min
anti-Nestin	Citrate buffer (pH 6.0)	Steaming 121 °C, 10 min
anti-PanCK	Target Retrieval Solution, Citrate (pH 6.1, 10×), S1699 (DAKO Agilent, Santa Clara, CA, USA)	1. Boiling 2. Waterbath, 95 °C, 20 min
anti-SOX2	Target Retrieval Solution, Citrate (pH 6.1, 10×), S1699 (DAKO Agilent, Santa Clara, CA, USA)	Steaming 125 °C, 4 min
anti-ZEB1	Citrate buffer (pH 6.0)	Steaming 121 °C, 10 min
anti-ZEB2	Citrate buffer (pH 6.0)	Steaming 121 °C, 10 min

**Table 3 cancers-16-00686-t003:** List of antibodies used for IHC of FFPE tissue sections from Panc1 and Panc89 Holo- or Paraclone cell-inoculated animals.

Primary Antibody	Secondary Antibody	Isotype
E-cadherin (1:100; clone NHC-38; DAKO Agilent, Santa Clara, CA, USA)	Dako REAL™ Detection System (Biotinylated goat anti-mouse/anti-rabbit immunoglobulins)	Mouse IgG1 (1:1484; 02-6100; Invitrogen via Thermo Fisher Scientific, Darmstadt, Germany)
L1CAM (1:1000; clone L1-11A, Peter Altevogt, German Cancer Research Center, Heidelberg, Germany)	Goat anti-mouse-biotin (1:200; LS-C149505; LS-Bio, Shirley, MA, USA)	Mouse IgG1 (1:228; 02-6100; Invitrogen via Thermo Fisher Scientific, Darmstadt, Germany)
Nestin (1:100; clone 10C2; Merck, Darmstadt, Germany)	Goat anti-mouse biotin (1:200; LS-C149505; LS-Bio, Shirley, MA, USA)	Mouse IgG1 (1:100; 02-6100; Invitrogen via Thermo Fisher Scientific, Darmstadt, Germany)
Pan-cytokeratin (PanCK) (12.7 µg/mL; clone AE1/AE3; DAKO Agilent, Santa Clara, CA, USA)	Goat anti-mouse biotin (1:200; LS-C149505; LS-Bio, Shirley, MA, USA)	Mouse IgG1 (1:80; 02-6100; Invitrogen via Thermo Fisher Scientific, Darmstadt, Germany)
SOX2 (1:20, polyclonal; Atlas Antibodies, Bromma, Sweden)	Goat anti-rabbit biotin (1:200; LS-C350860; LS-Bio, Shirley, MA, USA)	Rabbit polyclonal IgG (1:2000; ab37415; Abcam, Cambridge, UK)
ZEB1 (1:100; polyclonal; Atlas Antibodies, Bromma, Sweden)	Goat anti-rabbit biotin (1:200; LS-C350860; LS-Bio, Shirley, MA, USA)	Rabbit polyclonal IgG (1:2500; ab37415; Abcam, Cambridge, UK)
ZEB2 (1:100; polyclonal; Atlas Antibodies, Bromma, Sweden)	Dako REAL™ Detection System (Biotinylated goat anti-mouse/anti-rabbit immunoglobulins)	Rabbit polyclonal IgG (1:5000; ab37415; Abcam, Cambridge, UK)

**Table 4 cancers-16-00686-t004:** Frequency Score (= % positively stained cells) and intensity score (= intensity of staining) used for quantification of IHC analysis.

Frequency Score	Intensity Score
1—Negative/low (0–10%)	1—Negative/low
2—Intermediate (11–50%)	2—Intermediate
3—High (51–90%)	3—Strong
4—Strong (>90%)	–

## Data Availability

The datasets used and/or analyzed during the current study are available from the corresponding author on reasonable request. The RNA-seq data have been deposited at GEO under the access ID GSE241182. The data are available to the reviewers using the secure token ‘wbyxocoublsrxij’.

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
