# Peer review of "Epithelial and Mesenchymal-like Pancreatic Cancer Cells Exhibit Different Stem Cell Phenotypes Associated with Different Metastatic Propensities"

_cancers, 2024, doi:10.3390/cancers16040686_

Round 1

Reviewer 1 Report

Comments and Suggestions for Authors

The current manuscript by Philipp et al entitled “Epithelial and mesenchymal pancreatic cancer cells exhibit different stem cell phenotypes being associated with different metastatic propensity” explores the association between cancer stem cell (CSC) and non-CSC populations in pancreatic ductal adenocarcinoma (PDAC) with distinct epithelial-mesenchymal transition (EMT) phenotypes. The study suggests that these phenotypic differences have implications in the metastatic behavior, cell fate, and treatment response in PDAC.

The timeliness and significance of the manuscript are underscored by the persistent challenges in treating PDAC, possible correlated with some of the proposed issues. The potential contribution of the study lies in the discovery of specific EMT phenotypes, capable of characterizing distinct CSC cell populations, with different impacts on metastasis formation and treatment response in PDAC.

However, the manuscript is not yet suitable for publication in its current form, lacking the necessary depth to support relevant conclusions adequately. While the data displayed in this manuscript clearly emphasizes differences between the two cell lines in the study (Panc1 and Panc89), a significant gap exists in establishing disparities within the Holo- and Paraclones of both cell lines. In fact, authors claim Holoclones as the population with higher amount of CSC, however the presented data supporting this claim appears insufficient. Strengthening this conclusion would involve assessing a more comprehensive set of CSC markers.

In most of the examined cellular characteristics, the performance of PDAC cells appears linked to their cellular type and epithelial-mesenchymal transition (EMT) phenotype, rather than being correlated with the cancer stem cell (CSC) content across various clones (Holo- and Paraclones). It is then challenging to establish a clear correlation between these properties and the content and characteristics of distinct CSC populations. This emphasizes the need for a more defined characterization of the proposed CSC-EMT phenotypes and their impact.

Some figure legends are incomplete and hard to follow (eg. Figure2). 

Given these considerations, it is difficult to understand the main contribution of this study.

Comments on the Quality of English Language

-

Author Response

Dear reviewer,

we thank you very much for the rapid evaluation of our manuscript entitled “Epithelial and mesenchymal pancreatic cancer cells exhibit different stem cell phenotypes being associated with different metastatic propensity” and for the valuable suggestions to improve our manuscript.

According to your comments, we have prepared a revised version of the manuscript that you will find enclosed with this letter. All changes in the manuscript have been highlighted in red.

Below, we will explain point-by-point how the arguments and criticisms have been dealt with.

Request: While the data displayed in this manuscript clearly emphasizes differences between the two cell lines in the study (Panc1 and Panc89), a significant gap exists in establishing disparities within the Holo- and Paraclones of both cell lines. In fact, authors claim Holoclones as the population with higher amount of CSC, however the presented data supporting this claim appears insufficient. Strengthening this conclusion would involve assessing a more comprehensive set of CSC markers.

Answer: We thank the reviewer for pointing this out. As requested, we have thoroughly modified Figure 2 to better illustrate the differences regarding stemness between Panc1 versus Panc89 cells but also between Holo- and Paraclones of either cell line. Moreover, we have thoroughly rewritten the entire manuscript to better outline the differences between CSC- and non-CSC cell populations.

Request: In most of the examined cellular characteristics, the performance of PDAC cells appears linked to their cellular type and epithelial-mesenchymal transition (EMT) phenotype, rather than being correlated with the cancer stem cell (CSC) content across various clones (Holo- and Paraclones). It is then challenging to establish a clear correlation between these properties and the content and characteristics of distinct CSC populations. This emphasizes the need for a more defined characterization of the proposed CSC-EMT phenotypes and their impact.

Answer: As outlined above, we have thoroughly rewritten the entire manuscript to better outline the EMT- and CSC-differences between CSC- and non-CSC cell populations. In addition, we agree with the reviewer that isolating, expanding and characterizing Holo- and Paraclone populations from other PDAC cell lines is reasonable to substantiate our findings. In fact, we have already started single cell cloning of other PDAC cell lines. However, since we do not know whether and when we will have expanded different cell populations from these cell lines, integration of data from their characterization is beyond the scope of this paper. However, to consider this important point we have added the following sentence in the Conclusion on page 29: “However, being aware of that these findings have been revealed only with two PDAC cell lines and their related CSC and non-CSC variants, studies with Holo- and Paraclone cell variants of other PDAC cell lines have to be conducted to corroborate these results.”

Request: Some figure legends are incomplete and hard to follow (eg. Figure2). 

Answer: We apologize for these error. We have thoroughly checked and modified all figure legends (including those of Figure 2).

Thank you very much for your consideration and efforts!

Sincerely yours,

S. Sebens, PhD

Reviewer 2 Report

Comments and Suggestions for Authors

Philipp and colleagues present an interesting manuscript on the molecular and functional characterization of putative pancreatic ductal adenocarcinoma (PDAC) CSC-populations with more epithelial or more mesenchymal properties. The relatively unbiased approach followed to characterize PDAC populations, essentially based on phenotypes (in this case holoclones versus paraclones), is an alternative approach to classical enrichment/sorting strategies based on specific markers, which might introduce some form of bias. Authors are to be commended for the extensive in vitro and in vivo data concerning the profile of the different populations. However, the reported findings are somewhat heterogeneous and it is difficult, solely based on the behaviour of two PDAC cell lines (Panc1 and Panc89), to take major conclusions regarding the association between epithelial or more mesenchymal-like cancer cells, the different stem cell phenotypes and metastatic propensity. The major differences reported are actually between the two cell lines and not between the different phenotypic clones (holoclones, paraclones) of each cell line. In other words, it seems the malignant behaviour is more cell line specific and not so clearly dependent on the intrinsic heterogeneity within each cell line. The manuscript needs to be changed and improved in order to reflect this point. The work would probably become clearer and more consistent by analysing an enlarged panel of PDAC cell lines with distinct profiles (epithelial-like versus mesenchymal-like, as the title suggests) instead of focusing on clonal differences within each cell line. Specific comments are provided below.

Major comments

- Abstract section, line 46: “Single-cell cloning revealed CSC (Holoclone) and non-CSC (Paraclone) clones from the PDAC cell lines Panc1 and Panc89.” The fact that paraclones can give rise to other paraclone colonies in vitro already shows that they have some type of stemness potential. For instance, Panc89 paraclone-derived cells can still give rise to 70% of holoclones, are resistant to gemcitabine treatment, give rise to a significant number of tumours in vivo and still present high metastatic capacity, attributes of CSC-enriched tumours demonstrated for holoclones. As Authors recognize in the discussion “Of note, tumor formation of Panc1 and Panc89 Paraclone cells suggested that non-CSC variants can gain CSC properties, since both Paraclone variants were able to induce tumor formation…”. These supposed non-CSCs can have expression of other CSC markers (other than SOX2) that were not assessed. Ultimately, Authors do not provide an experiment formally demonstrating that these Paraclones are non-CSC populations. In the absence of, for instance, a serial xenotransplantation assay to demonstrate that these paraclones-derived cells might eventually lose their tumour-forming capacity, the Reviewer would advise against making such direct association (paraclones are non-CSC) so bluntly;

- From a content standpoint, the description of the results gets very confusing at times due to the attempt of detailing every possible comparison between the parental cell lines and the matching morphology-based clones, between each clone type, and between the two different cell lines. It is possible (and desirable) to simplify the description of the results, particularly avoiding the reference to small (probably biologically irrelevant) differences between samples, while keeping the major messages concerning the differences in phenotypes, allowing more straightforward reading;

- Results section, line 422: Authors state that Holoclone and Paraclone cell lines were generated via single-cell cloning. At least three clones of each were generated and used for the RNA-sequencing. Did the Authors always used the same clonal variant for the remaining experiments? If so, which one (clone 1, 2 or 3)? Or where these employed in a random fashion? If so, please define which clone was used for each experiment;

- Results section, Figure 1B: couldn`t Authors use 96-well plate single-cell dilution to derive each clone type (holoclone, paraclone) and to quantify the different types of colonies generated? This is a “one cell” colony formation assay. Why is there the need to go to 6-well plates with 400 cells/well?

- Results section, line 469: there is an almost 60% difference in the transcriptome between both parental PDAC cell lines. As stated above, it seems cell line intrinsic differences are more relevant towards defining the observed outcomes. Could the Authors discuss this?

- Results section, Figure 2B: the heatmap shows that there is not a good segregation of sample type. For instance, Panc89_holo1 and Panc89_para2 have a very different profile then the other two corresponding biological replicas. The same for Panc1_para3 that does not show the same trend concerning positive regulation of EMT and invasion. This also applies to stemness characteristics concerning Panc89_para2 and Panc1_para1. The description in the text (lines 508 to 525) should be simplified, but also needs to convey that there is a divergent pattern of association for some replicas;

- Results section, line 481: the relevance of using the “Mueller Plurinet” gene set is not clear. Authors should explain their rational for using this specific gene signature. It would be better to use a kind of stemness pan-signature derived from different studies or databases to minimize bias;

- Results section, Figure 4: Panc1 cells have a mesenchymal-like phenotype concerning the expression profile of specific markers (VIM, ZEB1 and L1CAM-positive, while E-cadherin-negative) and more invasive properties. However, they exhibit lower migration capacity, when compared to Panc89 cells. Do Authors have an explanation for this paradox?

- Results section, Figure 5: in line with the main criticism, the in vivo data shows that differences are mainly related with each cell line. Mice survival is mainly affected by Panc89 inoculation due to high tumour burden, much more than with Panc1. Even for the former, no significant differences in survival were observed between Holoclone and Paraclone cells. Organ dissemination is also clearly different between cell lines - Panc1 is more prone to stay in the pancreas and Panc89 is more prone to metastasize to the peritoneum. Paraclones inoculated mice do have a reduction in the total number of lesions (particularly for Panc1), but tumour area does not seem significantly different (statistics should be provided). In the case of Panc89 Paraclone inoculated mice, it led to the largest microscopic tumour areas observed. At the very least, results are not entirely concordant between the two cell lines for holoclone and paraclone behaviour, making it difficult to support the stated conclusions. These points must be taken into account in a reviewed version;

- Results section, Figure 5B: for the Panc1_holo inoculated mice, the graph shows 5/10 mice without macroscopic tumours, but the text refers that 7/10 animals had tumours. Please clarify this discrepancy; 

- Results section, lines 770-772: "Panc1 cell variants were not killed because of reduced health status due to the tumor burden, but very slow tumor progression terminating the experiment in the end without harming animals included." Panc1 mice did not present disease symptoms when they were euthanized? If so, why not extending the time of the experiment for Panc1 mice? What was the criteria for choosing the reported time-point (146 days)?

- Discussion section, lines 1038-1041: “Accordingly, Panc1 Holoclone cells exhibiting the strongest invasive potential using a mesenchymal invasion mode led to the highest number of metastatic lesions predominantly in the pancreas compared Panc1 Paraclones as well as epithelial Panc89 cell variants which formed less but larger tumors.” On the contrary, Panc89 cells seem clearly more metastatic, with a higher number of macroscopic and microscopic lesions in the peritoneum. Most Panc1_Holo microscopic lesions are located in the pancreas, which would be the organ of dissemination, showing that this cell line prefers the pancreatic niche. Please, discuss this;

- Conclusions section: Overall, the results do not support the claim that (Panc1) mesenchymal-like cancer cells have a higher propensity to spread and colonize secondary sites. Panc89 epithelial-like cells give rise to a higher number of macroscopic lesions, with a significantly higher median tumour area, and have increased metastatic ability originating a higher number of lesions in the peritoneum when compared to Panc1. Please, discuss this.

Minor comments

Introduction section, line 124: Reviewer advises the Authors not to use the terminology “mesenchymal Panc1 cells” or similar across the text. It conveys the wrong idea that these are stromal cells, while in fact the idea to be conveyed is that these are epithelial cancer cells with a mesenchymal-like phenotype;

Results section, Figure 2C: Reviewer advises Authors to change the colour scheme in this graphic, using the same colour type for each cell line (for instance, blue for Panc1 and green for Panc89) but with different tones according to clone, to more easily associate expression results to a specific cell line, like in Figure 3A;

Results section, Figure 2G: correct the text indications in the figures (Holoklon to holoclone, Paraklon to paraclone);

Results section, lines 642 and 643: “Next, it was analyzed whether these different cell growth abilities are related to different responses to cytostatic drugs.” Authors mean “Next, it was analyzed whether these different cell growth abilities influence the response to cytostatic drugs”.

Results section, Figure 3B: nuclei count analysis by Hoechst 33342 staining is not the most accurate readout for cell viability upon treatment. A live/dead staining or any other viability assay would be preferable;

Results section, Figure 4G: this assay is somewhat artificial and results are not in accordance with in vivo data (for instance, there is no difference in adhesion to mesothelial Met-5a cells, but Panc89 cells preferentially metastasize to the peritoneum). As it does not add additional information regarding metastatic propensity, the Reviewer would advise its removal.

Comments on the Quality of English Language

Minor typos to correct along the text, such as:

Introduction section, line 64: “and a still…“ instead of "and an still...";

Introduction section, line 66: rephrase sentence: “leaving palliative treatment as the remaining option.

Introduction section, line 68: rephrase sentence: “patients with liver or peritoneal metastases…”;

Introduction section, line 88: rephrase sentence: “to both self-renew and generate more differentiated cells”.

Author Response

Dear reviewer,

we thank you very much for the rapid evaluation of our manuscript entitled “Epithelial and mesenchymal pancreatic cancer cells exhibit different stem cell phenotypes being associated with different metastatic propensity” and for the valuable suggestions to improve our manuscript.

According to your comments, we have prepared a revised version of the manuscript that you will find enclosed with this letter. All changes in the manuscript have been highlighted in red.

Below, we will explain point-by-point how the arguments and criticisms have been dealt with.

Request: The work would probably become clearer and more consistent by analysing an enlarged panel of PDAC cell lines with distinct profiles (epithelial-like versus mesenchymal-like, as the title suggests) instead of focusing on clonal differences within each cell line. Specific comments are provided below.

Answer: We agree with the reviewer that isolating, expanding and characterizing Holo- and Paraclone populations from other PDAC cell lines is necessary to substantiate our findings. In fact, we have already started single cell cloning of other PDAC cell lines. However, since we do not know whether and when we will have expanded different cell populations from these cell lines, integration of data from their characterization is beyond the scope of this paper. However, to consider this important point we have added the following sentence in the Conclusion on page 29: “However, being aware of that these findings have been revealed only with two PDAC cell lines and their related CSC and non-CSC variants, studies with Holo- and Paraclone cell variants of other PDAC cell lines have to be conducted to corroborate these results.” Furthermore, we have thoroughly rewritten the entire manuscript to better outline the EMT- and CSC-differences between CSC- and non-CSC cell populations. This implied also a thorough review of the wording used for the conclusions. 

Major comments

Request: Abstract section, line 46: “Single-cell cloning revealed CSC (Holoclone) and non-CSC (Paraclone) clones from the PDAC cell lines Panc1 and Panc89.” The fact that paraclones can give rise to other paraclone colonies in vitro already shows that they have some type of stemness potential. For instance, Panc89 paraclone-derived cells can still give rise to 70% of holoclones, are resistant to gemcitabine treatment, give rise to a significant number of tumours in vivo and still present high metastatic capacity, attributes of CSC-enriched tumours demonstrated for holoclones. As Authors recognize in the discussion “Of note, tumor formation of Panc1 and Panc89 Paraclone cells suggested that non-CSC variants can gain CSC properties, since both Paraclone variants were able to induce tumor formation…”. These supposed non-CSCs can have expression of other CSC markers (other than SOX2) that were not assessed. Ultimately, Authors do not provide an experiment formally demonstrating that these Paraclones are non-CSC populations. In the absence of, for instance, a serial xenotransplantation assay to demonstrate that these paraclones-derived cells might eventually lose their tumour-forming capacity, the Reviewer would advise against making such direct association (paraclones are non-CSC) so bluntly.

Answer: Besides tumorigenicity assays, colony formation assays are well established readout procedures to assess the self-renewal capacity of tumor cells and the differentiation of CSC versus non-CSC populations according to the colony morphology has been widely acknowledged. Furthermore, several studies have already shown that colony types can switch their phenotype being in line with the current understanding of CSC as a plastic cell population which can gain and lose CSC properties. We have outlined this important aspect in more detail in the introduction on page 3 and 4. Furthermore, we clearly outline not only in the Material & Methods section but also in the Results section that a very low number of cells (1x104) was used for the tumorigenicity and metastasis assay (please see page 21)

Request: From a content standpoint, the description of the results gets very confusing at times due to the attempt of detailing every possible comparison between the parental cell lines and the matching morphology-based clones, between each clone type, and between the two different cell lines. It is possible (and desirable) to simplify the description of the results, particularly avoiding the reference to small (probably biologically irrelevant) differences between samples, while keeping the major messages concerning the differences in phenotypes, allowing more straightforward reading;

Answer: We agree with the reviewer that the description of the results was not sufficiently clear and comprehensible to the readership. Therefore, we have thoroughly revised the entire results section, focusing more on the comparison between parental cell lines but also between holo- versus paraclone cells (= CSC- versus non-CSC populations). As part of this careful revision, we also modified the presentation of the data in Figure 2.

Request: Results section, line 422: Authors state that Holoclone and Paraclone cell lines were generated via single-cell cloning. At least three clones of each were generated and used for the RNA-sequencing. Did the Authors always used the same clonal variant for the remaining experiments? If so, which one (clone 1, 2 or 3)? Or where these employed in a random fashion? If so, please define which clone was used for each experiment;

Answer: We thank the reviewer for asking this important point. In the revised manuscript we provide information on the clones of each cell variant used throughout the study, please see page 5 and page 7. Thus, on page 5 we write: “All experiments of this study were performed with defined Holo- and Paraclone cells isolated from both Panc1 cell clones (clone 9, clone5, respectively) and Panc89 cell clones (clone 4, clone 3, respectively).”

Request: Results section, Figure 1B: couldn`t Authors use 96-well plate single-cell dilution to derive each clone type (holoclone, paraclone) and to quantify the different types of colonies generated? This is a “one cell” colony formation assay. Why is there the need to go to 6-well plates with 400 cells/well?

Answer: Single cell cloning and CFA are performed with different goals. While a single-cell cloning does not only aim at isolation but also expansion of colonies generated from a single cell, a colony formation assay aims at counting a reliable amount of colonies from single cells of a defined cell population. These different aims determine the use of different well formats. We have slightly modified description of single cell cloning and CFA (please see page 4 and 5).

Request: Results section, line 469: there is an almost 60% difference in the transcriptome between both parental PDAC cell lines. As stated above, it seems cell line intrinsic differences are more relevant towards defining the observed outcomes. Could the Authors discuss this?

Answer: In course of our careful revision, we have rewritten the results (description of Figure 2 please see pages 12-14) and have modified presentation of data in Figure 2 to outline cell line specific differences but also the clearly prevalent differences between the respective Holo- and Paraclone cells. Besides, we speculate on how the differences between the two PDAC cell lines determine the findings observed in this study, please see Discussion particularly on page 29.

Request: Results section, Figure 2B: the heatmap shows that there is not a good segregation of sample type. For instance, Panc89_holo1 and Panc89_para2 have a very different profile then the other two corresponding biological replicas. The same for Panc1_para3 that does not show the same trend concerning positive regulation of EMT and invasion. This also applies to stemness characteristics concerning Panc89_para2 and Panc1_para1. The description in the text (lines 508 to 525) should be simplified, but also needs to convey that there is a divergent pattern of association for some replicas;

Answer: As already outlined above, we agree with the reviewer that the presentation of the results was not sufficiently clear and comprehensible for the reader. Therefore, we have thoroughly revised description (see pages 12-14) and presentation of findings of Figure 2.

Request: Results section, line 481: the relevance of using the “Mueller Plurinet” gene set is not clear. Authors should explain their rational for using this specific gene signature. It would be better to use a kind of stemness pan-signature derived from different studies or databases to minimize bias;

Answer: The “Mueller Plurinet” gene set was used because our bioinformaticians have a very good and long-standing experience with it in order to particularly reflect biological differences. Using this gene set our analysis revealed a higher gene activity in stemness related genes in parental Panc89 cells compared to parental Panc1 cells as well as in the two holoclonal populations compared to the paraclonal populations. Supporting our experience, these findings were highly consistent with the functional findings.

Request: Results section, Figure 4: Panc1 cells have a mesenchymal-like phenotype concerning the expression profile of specific markers (VIM, ZEB1 and L1CAM-positive, while E-cadherin-negative) and more invasive properties. However, they exhibit lower migration capacity, when compared to Panc89 cells. Do Authors have an explanation for this paradox?

Answer: As requested, we discuss this point in the Discussion on page 28. Here, we write “Despite the more pronounced invasion ability, Panc1 cell variants were characterized by slower cell migration compared to Panc89 cell variants. This observation might be explained by the fact that Panc1 cell variants exhibited a higher expression of proteases such as matrix metalloproteinase (MMP)-2 and MMP-9 compared to Panc89 cell variants (data not shown). Both MMPs enable degradation of collagen, being a major structural component of basement membranes and matrigel which is used in the invasion assays, and thus facilitating cellular invasion (99,100).”

Request: Results section, Figure 5: in line with the main criticism, the in vivo data shows that differences are mainly related with each cell line. Mice survival is mainly affected by Panc89 inoculation due to high tumour burden, much more than with Panc1. Even for the former, no significant differences in survival were observed between Holoclone and Paraclone cells. Organ dissemination is also clearly different between cell lines - Panc1 is more prone to stay in the pancreas and Panc89 is more prone to metastasize to the peritoneum. Paraclones inoculated mice do have a reduction in the total number of lesions (particularly for Panc1), but tumour area does not seem significantly different (statistics should be provided). In the case of Panc89 Paraclone inoculated mice, it led to the largest microscopic tumour areas observed. At the very least, results are not entirely concordant between the two cell lines for holoclone and paraclone behaviour, making it difficult to support the stated conclusions. These points must be taken into account in a reviewed version;

Answer: As requested we have modified the description of the entire results, including those of Figure 5. In the revised version, we have also linked these findings better to the findings of the in vitro characterization. Moreover, we have carefully modified the description of the results (please see page 22) as well as their discussion (please see page 27-30).

Request: Results section, Figure 5B: for the Panc1_holo inoculated mice, the graph shows 5/10 mice without macroscopic tumours, but the text refers that 7/10 animals had tumours. Please clarify this discrepancy. 

Answer: We apologize for having overseen this mistake. We have corrected the information accordingly on page 22.

Request: Results section, lines 770-772: "Panc1 cell variants were not killed because of reduced health status due to the tumor burden, but very slow tumor progression terminating the experiment in the end without harming animals included." Panc1 mice did not present disease symptoms when they were euthanized? If so, why not extending the time of the experiment for Panc1 mice? What was the criteria for choosing the reported time-point (146 days)?

Answer: Panc89 cell variants inoculated mice were sacrificed because of reduced health status due to high tumor burden. In contrast, animals inoculated with Panc1 cell variants showed very slow tumor progression leading to only small lesions which did not further grow out even after 5 months (= 146 days). Therefore, it was decided to terminate experiment prematurely. We have outlined this information in the respective Material & Methods section on page 9 and have changed the information in the results section on page 22: “Mice inoculated with Panc89 cell variants were sacrificed when the health status was impaired due to a high tumor burden demanding removal of the animals from the experiment. In contrast, animals inoculated with Panc1 cell variants showed very slow tumor progression leading to only small lesions which did not further grow out even after 5 months (= 146 days). Therefore, it was decided to terminate experiment prematurely.”

Request: Discussion section, lines 1038-1041: “Accordingly, Panc1 Holoclone cells exhibiting the strongest invasive potential using a mesenchymal invasion mode led to the highest number of metastatic lesions predominantly in the pancreas compared Panc1 Paraclones as well as epithelial Panc89 cell variants which formed less but larger tumors.” On the contrary, Panc89 cells seem clearly more metastatic, with a higher number of macroscopic and microscopic lesions in the peritoneum. Most Panc1_Holo microscopic lesions are located in the pancreas, which would be the organ of dissemination, showing that this cell line prefers the pancreatic niche.

Answer: As requested, we discuss the findings of the metastatic patterns in more detail in the Discussion on pages 28 and 29. Here we write: “Although Panc1 Holoclone cells also led to a higher number of microscopic lesions compared Panc89 Holoclone cells, inoculation of Panc89 Holoclone cells yielded the most pronounced overall tumor burden with the highest number macroscopic tumors of large sizes compared to all other cell variants being well in line with the highest proliferative and self-renewal ability of these cells observed in vitro. The fact that inoculation of Panc1 and Panc89 Paraclone cells led to tumor formation, albeit to a lesser extent, supports the view of a high plasticity in these cell populations implying that non-CSCs can gain CSC properties. Notably, despite this possible gain of CSC properties in Panc1 and Panc89 Paraclone cells, overall expression of CSC-markers was rather low in these tumors. Thus, further studies are needed to investigate the role of other CSC markers and factors, e.g. provided by the tumor microenvironment (see above), that determine phenotypic switching of non-CSCs and CSCs. In contrast to Panc1 tumors, which predominantly occurred in the pancreas, tumors formed by Panc89 cell variants predominantly manifested in the peritoneum. Previous studies have demonstrated that PDAC patients with peritoneal recurrences exhibit significantly shorter disease-free survival and worse overall prognosis compared to PDAC patients with e.g. pulmonary recurrences (7), which is consistent with the shorter survival times of animals being inoculated with peritoneal metastasis forming Panc89 cell variants. Since adhesion assays did not reveal any conclusive differences of PDAC cell adhesion to organ specific endothelial cells between Holo- and Paraclone populations of either cell line, other factors seemed to be more crucial in determining the tumor manifestation patterns. The fact that mesenchymal-like Panc1 cells are derived from the primary tumor, together with their gene activity associated with EMT and invasion, suggests that these cells have undergone EMT to leave the primary tumor. However, whether these cells would ever have been able to form metastases in this patient remains unsolved. Furthermore, it may explain, why most Panc1 Holoclone tumors were found in the pancreas, indicating that these cells are still optimally adapted to their original tissue. In contrast, epithelial Panc89 cells originate from a lymph node metastasis, thus these cells have proven their disseminating potential to leave the primary tumor (either after EMT using a mesenchymal-like invasion mode or while maintaining an epithelial/hybrid cell stage using a cluster-like mode) and to grow out as metastasis at a secondary site (e.g., after MET). Thus, they have managed to survive all steps of the metastatic cascade including adaptation to novel microenvironments. It can be speculated whether the patient, from which Panc89 cells were derived from, had developed peritoneal metastases in course of the disease which would be consistent with tumor manifestation in our in vivo model. Furthermore, it would be of great interest to investigate whether primary tumors of PDAC patients who developed peritoneal metastases contain more epithelial CSCs, and primary tumors of patients with other metastatic sites show a more mesenchymal-like CSC phenotype.”

Request: Conclusions section: Overall, the results do not support the claim that (Panc1) mesenchymal-like cancer cells have a higher propensity to spread and colonize secondary sites. Panc89 epithelial-like cells give rise to a higher number of macroscopic lesions, with a significantly higher median tumour area, and have increased metastatic ability originating a higher number of lesions in the peritoneum when compared to Panc1. Please, discuss this.

Answer: Based on the revised discussion we have also carefully modified the conclusion chapter (please see page 30) as follows: “In summary, these results support the view that mesenchymal-like CSC have a strong propensity to colonize secondary sites and to form higher number of (small) tumoral lesions which however does not ultimately lead to fast disease progression associated with short survival. In contrast, an epithelial CSC-phenotype seemed to be associated with slow cell invasion but the concomitant advantage of rapid tumor outgrowth, resulting in a fast increase of a life-threatening tumor burden (exemplified by the highest number of macroscopic tumors and shorter survival). Overall, our data support the view that different CSC-phenotypes exist in PDAC which are associated with distinct EMT-phenotypes of PDAC cells essentially determining PDAC cell fate and function as well as treatment responses. However, being aware of that these findings have been obtained only with two PDAC cell lines and their related CSC and non-CSC variants, studies with Holo- and Paraclone cell variants derived from other PDAC cell lines have to be conducted to corroborate these results.”

Minor comments

Request: Introduction section, line 124: Reviewer advises the Authors not to use the terminology “mesenchymal Panc1 cells” or similar across the text. It conveys the wrong idea that these are stromal cells, while in fact the idea to be conveyed is that these are epithelial cancer cells with a mesenchymal-like phenotype.

Answer: We thank the reviewer for this valuable comment and use “mesenchymal-like cells” throughout the entire revised manuscript.

Request: Results section, Figure 2C: Reviewer advises Authors to change the colour scheme in this graphic, using the same colour type for each cell line (for instance, blue for Panc1 and green for Panc89) but with different tones according to clone, to more easily associate expression results to a specific cell line, like in Figure 3A.

Answer: As requested we have modified the coloring of the cell type and have used the same color type for each cell line in Figure 2A and 2C.

Request: Results section, Figure 2G: correct the text indications in the figures (Holoklon to holoclone, Paraklon to paraclone).

Answer: We thank the reviewer for this attentive notice. We have corrected the figure accordingly.

Request: Results section, lines 642 and 643: “Next, it was analyzed whether these different cell growth abilities are related to different responses to cytostatic drugs.” Authors mean “Next, it was analyzed whether these different cell growth abilities influence the response to cytostatic drugs``.

Answer: As requested we have modified the sentence accordingly.

Request: Results section, Figure 3B: nuclei count analysis by Hoechst 33342 staining is not the most accurate readout for cell viability upon treatment. A live/dead staining or any other viability assay would be preferable;

Answer: Since we have used an automated microscopy device for determination of treatment responses, we have decided to use Hoechst 33342 staining to determine the total number of still adherent cells as a thorough testing revealed this as the most reliable readout procedure. During tests with live/dead cell staining it was noted that dead/apoptotic cells detach from the cell plate and further disintegrate into apoptotic bodies which are not reliably counted. We have modified the methods description on page 8 accordingly: “To assess treatment responses of the different PDAC cell variants, the total number of adherent cells, being left untreated or exposed to treatment with cytostatic drugs for 72h, was determined. For this purpose,…”

Request: Results section, Figure 4G: this assay is somewhat artificial and results are not in accordance with in vivo data (for instance, there is no difference in adhesion to mesothelial Met-5a cells, but Panc89 cells preferentially metastasize to the peritoneum). As it does not add additional information regarding metastatic propensity, the Reviewer would advise its removal.

Answer: We agree with the reviewer that adhesion properties assessed in the adhesion assays do not explain the different metastatic manifestations in vivo. Therefore, we have shortened and sharpened the description of the results (page 20) and have omitted Figure 4G. The results are now shown in the novel Supplementary Figure 2.

Request: Minor typos to correct along the text, such as:

Introduction section, line 64: “and a still…“ instead of "and an still...";

Introduction section, line 66: rephrase sentence: “leaving palliative treatment as the remaining option.

Introduction section, line 68: rephrase sentence: “patients with liver or peritoneal metastases…”;

Introduction section, line 88: rephrase sentence: “to both self-renew and generate more differentiated cells”.

Answer: All minor typos have been corrected.

Thank you very much for your consideration and efforts!

Sincerely yours,

S. Sebens, PhD

Round 2

Reviewer 1 Report

Comments and Suggestions for Authors

In a broad perspective, the second version of the manuscript “Epithelial and mesenchymal pancreatic cancer cells exhibit different stem cell phenotypes being associated with different metastatic propensity” by Philipp et al. has substantially improved readability, facilitating a clearer understanding of the presented findings and the ability to draw conclusions, when compared to the initial version.

The clarity extends to both the presentation of results and the derivation of conclusions, as well as the graphical elements, including figures and their respective legends (notably Figure 2).

Regarding some issues, the authors claim that they observed distinct gene expressions among the clone variants of PANC1 and PANC89 cell lines (Supplementary Figure 1E). While I acknowledge these differences among the PANC89 clone variants, I hold a differing opinion regarding the clarity of these differences in the case of the PANC1 clones.

Additionally, the authors state that the parental PANC1 cell population consists of cells exhibiting both NES and ZEB-2. However, the chosen image (Figure 2.G) may not be the most illustrative of this result, as the expression of ZEB-2 in this specific condition is not clear.

Also, the supplementary figures lack corresponding legends, impeding the reader's complete understanding. Providing additional information regarding these figures would enhance their interpretability.

Furthermore, as a constructive indication, I would suggest adjusting the colors/symbols in Figure 5.A, to facilitate figure’s interpretation and identification of cell variants under discussion.

Several minor issues related to the sentence construction have been identified, such as:

a.       The use of "applies" instead of "applied" on line 900.

b.       A potential sentence error on line 1014, where the phrase "animals being inoculated with PAN89 cell variants forming peritoneal metastasis" may need clarification.

Overall, this revised manuscript effectively guides the reader through the authors' thought process, aligning them with the main objectives and highlighting the paper's contribution. In fact, the exploration of differences in the CSC populations of PDAC cell lines holds significant implications for cancer treatment strategies, that need to be revised.

Comments on the Quality of English Language

-

Author Response

Dear Reviewer,

thank you very much again for the rapid evaluation of our manuscript entitled “Epithelial and mesenchymal-like pancreatic cancer cells exhibit different stem cell phenotypes being associated with different metastatic propensity” and for the valuable suggestions  to further improve our manuscript.

According to your comments, we have prepared a further revised version of the manuscript that you will find enclosed with this letter. All new changes in the manuscript have been highlighted in red.

Below, we will explain point-by-point how the arguments and criticisms have been dealt with.

First of all we thank you for acknowledging the comprehensive improvements of our manuscript in terms of readability and comprehensibility.

Request: Regarding some issues, the authors claim that they observed distinct gene expressions among the clone variants of PANC1 and PANC89 cell lines (Supplementary Figure 1E). While I acknowledge these differences among the PANC89 clone variants, I hold a differing opinion regarding the clarity of these differences in the case of the PANC1 clones.

Answer: As the suggestions of the reviewer have contributed to a clear improvement of manuscript, we also appreciate his opinion on the differences between Panc1 Holo- and Paraclones. In fact, both clones do not that differ with respect to EMT characteristics (than the two Panc89 cell variants do), however there are several clear differences between the two clone variants which are:

  1. Colony formation ability: Panc1 Holoclone cells form clearly more holoclones than paraclones while Paraclone cells most exclusively form paraclones (Figure 1B).
  2. Gene expression Analyses: In the PCA (Figure 2A), Holoclone cells are markedly different from Paraclone cells which is substantiated by the GSVA (FIgure 2B). Furthermore, throughout all transcriptom analyses Panc1 Holoclone cells have been characterized by a more pronounced CSC phenotype than Panc1 Paraclones cells (being in line with results regarding colony formation in vitro and tumor and metastases formation in vivo).
  3. Cell growth behavior: Panc1 Holoclone cells grow faster compared to Paraclone cells (Figure 3A)
  4. Tumor and metastases formation: Panc1 Holoclone cells formed more and larger tumors/metastases than Panc1 Paraclone cells.

Thus, based on these results we are convinced of our results demonstrating clear differences between Panc1 Holo- and Paraclone cells.

Request: Additionally, the authors state that the parental PANC1 cell population consists of cells exhibiting both NES and ZEB-2. However, the chosen image (Figure 2.G) may not be the most illustrative of this result, as the expression of ZEB-2 in this specific condition is not clear.

Answer: As shown in Figure 2F, ZEB2 RNA expression is very low in parental Panc1 cells and only slightly higher than those of Panc1 Paraclone cells. In line with these qPCR results, ZEB1 staining is very faint in parental cells and only marginally better detectable than those of Panc1 Paraclone cells (Figure 2G). We have modified the respective results description as follows: „Finally, double IFS was performed to confirm gene expression differences on protein level and to particulary discriminate Holo- and Paraclone cells from each other.“ Furthermore, it can be speculated that ZEB2 expression which can be regarded as a marker for Panc1 Holoclone cells is diminished in the presence of other clone variants in the parental population.

Request: Also, the supplementary figures lack corresponding legends, impeding the reader's complete understanding. Providing additional information regarding these figures would enhance their interpretability.

Answer: We are surprised about this comment as we have submitted revised legends to all supplementary figures. Along with the revised manuscript we again upload the supplementary figure legends.

Request: Furthermore, as a constructive indication, I would suggest adjusting the colors/symbols in Figure 5.A, to facilitate figure’s interpretation and identification of cell variants under discussion.

Answer: As suggested we have modified Figure 5A to facilitate identification of the cell variants.

Request: The use of "applies" instead of "applied" on line 900.

Answer: We have removed the “is” and modified the sentence as follows: “Besides, CSCs and EMT are both linked to cancer progression and early metastasis, which also applies to PDAC.”

Request: A potential sentence error on line 1014, where the phrase "animals being inoculated with PAN89 cell variants forming peritoneal metastasis" may need clarification.

Answer: We agree with the reviewer that this sentence was somehow misleading. We have modified the sentence as follows: „[…] which is consistent with the shorter survival times of animals being inoculated with Panc89 cell variants that formed peritoneal metastasis.“

Thank you very much for your consideration and efforts!

Sincerely yours,

S. Sebens, PhD